# The most polyphagous insect herbivore? Host plant associations of the Meadow spittlebug, *Philaenus spumarius* (L.)

**Vinton Thompson[1]\*, Claire Harkin[2,3]\*, Alan J. A. Stewart[2]\***

**1** Division of Invertebrate Zoology, American Museum of Natural History, New York, New York, United States of America, **2** School of Life Sciences, University of Sussex, Falmer, Brighton, East Sussex, United Kingdom, **3** Ancient Tree Forum, London, United Kingdom

\* vthompson@mcny.edu (VT); claire.harkin@ancienttreeforum.org.uk (CH); a.j.a.stewart@sussex.ac.uk (AJAS)

**Data Availability Statement:** All relevant data are within the paper.

**Funding:** The BRIGIT project was funded by UK Research and Innovation (https://www.ukri.org/)

## Abstract

A comprehensive list of all known host plant species utilised by the Meadow Spittlebug (*Philaenus spumarius* (L.)) is presented, compiled from published and unpublished sources. *P. spumarius* feeds on 1311 host plants in 631 genera and 117 families. This appears, by a large margin, to be the greatest number of host species exploited by any herbivorous insect. The Asteraceae (222 species) and Rosaceae (110) together account for 25% of all host species. The Fabaceae (76) and Poaceae (73), are nearly tied for third and fourth place and these four families, combined with the Lamiaceae (62), Apiaceae (50), Brassicaceae (43) and Caprifoliaceae (34), comprise about half of all host species. Hosts are concentrated among herbaceous dicots but range from ferns and grasses to shrubs and trees. *Philaenus spumarius* is an "extreme polyphage", which appears to have evolved from a monophage ancestor in the past 3.7 to 7.9 million years. It is also the primary European vector of the emerging plant pathogen *Xylella fastidiosa*. Its vast host range suggests that it has the potential to spread *X. fastidiosa* among multiple hosts in any environment in which both the spittlebug and bacterium are present. Fully 47.9% of all known hosts were recorded in the *Xylella*-inspired BRIGIT citizen science *P. spumarius* host survey, including 358 hosts new to the documentary record, 27.3% of the 1311 total. This is a strong demonstration of the power of organized amateur observers to contribute to scientific knowledge.

## Introduction

The Meadow spittlebug *Philaenus spumarius* (L.) (Hemiptera: Aphrophoridae) is one of the world's most widespread and abundant insects (Fig 1). It is also the major European vector of *Xylella fastidiosa* Wells et al., an emerging bacterial plant pathogen that threatens crops as diverse as grapes, almonds, citrus and olives [1–4]. Despite its importance, the last comprehensive review of *P. spumarius* host plants is almost 70 years old [5]. Concern with *X. fastidiosa* has led to a proliferation of recent studies adding new *P. spumarius* hosts (e.g. [6–11]), including the BRIGIT citizen scientist initiative in Britain that enlisted amateurs to identify *P. spumarius* host plants [12].

through the Strategic Priorities Fund, by a grant from the Biotechnology and Biological Sciences Research Council (https://www.ukri.org/councils/bbsrc/), with support from the UK Department for Environment, Food and Rural Affairs and the Scottish Government to A.J.A.S and C.H. The funders had no role in study design, data collection and analysis, decision to publish, or preparation of the manuscript.

**Competing interests:** The authors have declared that no competing interests exist.

The present work has multiple aims. First, we bring together disparate sources in a publicly available, comprehensive, documented compilation of *P. spumarius* hosts. This will be of use to investigators studying the spread of *X. fastidiosa* and broader biological phenomena, such as the evolution of xylem feeding insects and the evolution of extreme polyphagy. Second, we analyse the broad patterns of *P. spumarius* host exploitation, place it in the context of other highly polyphagous insects, and examine the implications of host patterns for *X. fastidiosa* spread. Third, we suggest some useful guidelines for future surveys of spittlebug host plants. Finally, we assess the efficacy of the BRIGIT project in corroborating existing host records and establishing new ones. We believe this is the first time a citizen science project of this scope has been tested against the historical record.

*Philaenus spumarius* first became a major subject of study in the late 1940s and early 1950s in the United States and Canada for its role as a serious non-native pest of alfalfa (*Medicago sativa*) and other forage legumes [5]. From about 1960 to 2010 it was primarily of interest to entomologists and ecologists working on insect population dynamics and energetics [13–16] and to researchers interested in its remarkable colour polymorphism [17–20]. More recently, it has attracted considerable attention from applied entomologists and pest management scientists, after it was shown to be the primary European vector of *X. fastidiosa*, a pathogen that causes Olive Quick Decline Syndrome (OQDS), which has devastated olive groves in the Apulia region of Italy [1, 21]. This threat, along with the recent association of *P. spumarius* with Almond leaf scorch disease (ALSD) in the Alicante and Balearic Islands regions of Spain [22, 23], has generated intense research activity focused on the meadow spittlebug, reflected in an exponential recent rise in publications and citations (175 papers and 2,699 citations since 2010; Web of Science, accessed 21 May 2023). *Xylella fastidiosa* was introduced from the New World to Europe relatively recently [23]. It has now been detected in over 500 plant species, including crops, ornamentals and trees, across many plant families [24]. To predict which agricultural and natural plant species and ecosystems are at risk, it is crucial to understand the host range of the vector.

Information on *P. spumarius* host plants is scattered across many languages and continents, from the primary scientific literature to amateur entomology publications, agricultural tracts and unpublished sources. Schmidt [25, 26], apparently an amateur, published the earliest extensive host lists in 1914, totaling 137 species, in a German civic booster journal promoting the progress of science and industry. The next major contributions came from Hawaii [27, 28],

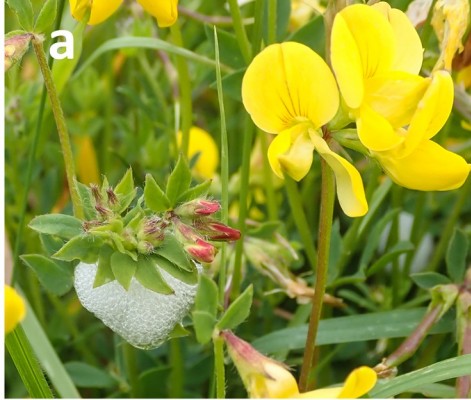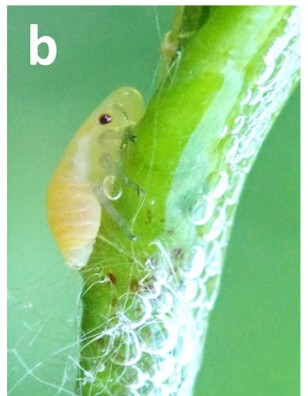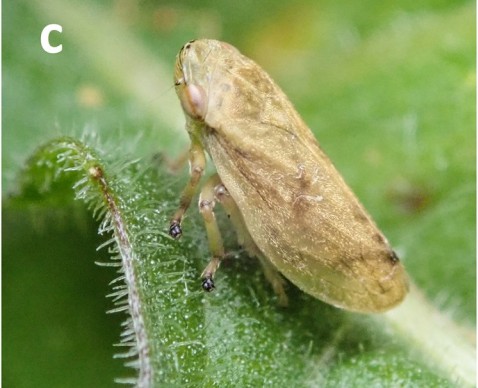

**Fig 1. The meadow spittlebug *Philaenus spumarius* (L.).** a) Frothy spittle mass on one of its common host plants, the legume *Lotus corniculatus*, bird's-foot trefoil. The nymph is hidden within. b) Spittle foam removed to expose nymph. The dark red dot is its right eye. Note the swollen sucking pump anterior to the eyes. c) Adult *P. spumarius*. This is the most common of several colour forms. Photographs by Claire Harkin.

following the introduction and proliferation of *P. spumarius* on Big Island in the 1940s. This work was followed shortly by the 1950 publication of 97 California hosts observed by DeLong and Severin [29] in the course of the first experiments demonstrating that *P. spumarius* and other spittlebugs can transmit what is now known to be *X. fastidiosa*. A series of host lists in American PhD theses [30–33] culminated in 1954 in the 383 species compilation of Weaver & King [5]. Metcalf's encyclopedic catalog of the Aphrophoridae [34] includes annotated references to host information in *P. spumarius* publications through 1955, a very useful source of otherwise obscure observations. In 1965 Noury [35] compiled a list of about 167 hosts from France and Middle and Northern Europe. Halkka et al. [36] provide a list of 165 hosts based on 1960s field observations in Finland. A 1993 PhD thesis by Booth [37] includes records of 90 hosts in natural areas in Wales and New Zealand. Jennifer Owen [38] and Denis Owen [39] record observing 143 host species in 47 plant families in their late twentieth century English suburban garden, detailed in notes that we recovered from J. Owen's papers in the Leicester Museum & Art Gallery. More recently, concern over *X. fastidiosa* has stimulated a golden age of *P. spumarius* host studies in the Mediterranean region (references below). We have collected, evaluated and distilled these sources and all others available to us into a single, readily accessible summary table in searchable format.

## Methods

### Sources of information

This work is based on the published literature, both formal and informal, host records associated with museum specimens, personal observations, private communications from colleagues, and the BRIGIT citizen science effort carried out in Britain. The first four sources are encompassed in an unpublished world database of spittlebug host plants built and maintained by VT. Ironically, it contains most published host records for most spittlebugs (Hemiptera: Cercopoidea) except for *P. spumarius*, which was initially exempted from full incorporation because the number of host records was enormous and because there seemed to be little point in recording the hosts of an insect that appeared to feed on almost any available plant. That changed with increasing concern about the role of *P. spumarius* as a *X. fastidiosa* vector. Thousands of *P. spumarius* records have been added, increasing the database from about 5,000 to 9,000 records over the past two years. Even with these additions it is not comprehensive for *P. spumarius*, missing for example, all of the BRIGIT citizen science records reported here, which are in a separate database maintained by CH and AJAS. The hosts presented here include all known to us as of May 2023.

The BRIGIT project [12] ran from 2019 to 2021 with the aim of improving surveillance and response capacity for *X. fastidiosa* should it be introduced into the UK. A key objective of the project was to develop a greater understanding of the distribution and host plant preferences of *P. spumarius*, identified as the primary vector of the bacterium in Europe [40]. Citizen scientists were encouraged to submit sightings via a national website for natural history observations or a bespoke portal, supported by identification aids on the BRIGIT website.

### Sources of ambiguity

**Anomalies in Weaver & King.** Prior to this work, Weaver & King's 1954 compilation [5] has been the primary source of *P. spumarius* host records. It originated as a shorter host list in a preceding thesis by King [30]. Ostensibly, it is an amalgam of 26 cited sources, combined with personal observations in Ohio by Weaver and King themselves. We have tracked down and examined all of the original sources cited. This revealed several anomalies. For example, Weaver and King state that all the records in their compilation are for nymphal hosts and go

on to say that they omitted the long 1950 California list compiled by DeLong & Severin [29] because those authors did not distinguish between nymphal and adult hosts. Nevertheless, they still cite four of the 97 hosts reported in that work, while omitting the rest. They list hosts from Osborn's 1916 work [41] without reservation, though Osborn too omits information on whether his observations are for nymphs or adults. Unlike Osborn, DeLong & Severin provide dates and locales for their observations. All were made in Alameda County, on the sunnier, warmer side of San Francisco Bay, in April or the first three weeks of May, or in San Francisco, on the foggier, cooler side of the bay, in April, May or the first week of July. In these areas, for the periods in question, most or all *P. spumarius* individuals are still in the nymphal stage (VT observations), so it may be reasonably inferred that DeLong & Severin observed nymphs for all listed hosts.

Weaver and King also inexplicably omit about two dozen nymphal hosts listed in Teller [32], while incorporating 33 others. Likewise, their treatment of Marshall [33] is problematic. They include most of his records, despite the fact that he does not specify life stage, but attribute one host record to him that is not in his work (*Rumex acetosa*). They also attribute two of Marshall's records (*Pinus strobus* and *Rumex occidentalis*) to Krauss [42], making it appear that they came from Hawaii rather than New York State. In another oddity, they misattribute a work by Davis [27] to the authors of a preceding work by Davis & Mitchell [28] and get the title of the paper by Davis wrong. These mistakes have the look of clerical errors. In the case of Licent [43], they include three of the hosts listed, but omit three others, perhaps a case of oversight, since the three unlisted hosts are in a different section of a very long treatise.

More disconcertingly, and for no obvious reason, Weaver and King omit 39 hosts out of 85 listed in Schmidt's second 1914 paper [25] and misstate the date of that publication as 1915. Given these anomalies and inconsistencies, we have included only hosts that could be corroborated in the original cited works and incorporate, where appropriate, hosts listed in the original sources that Weaver and King omitted. We include the host records Weaver and King report as their own observations.

**Sitting records versus host records.**   We distinguish among records for nymphal hosts, adult hosts and plants that host both stages. Nymphal host records are unambiguous because nymphs (Fig 1b) cannot produce spittles (Fig 1a) without feeding. Adult records are more problematic. Adult *P. spumarius* (Fig 1c) are active insects and move about from plant to plant. Unless accompanied by direct observation of feeding in the form of regularly expelled droplets of excreta, adult records may represent "sitting" rather than feeding individuals. This produces inevitable "noise" in adult host records. To minimize this noise, we have omitted instances in which adult host records are clearly unreliable, including those based on single collected specimens. Nevertheless, some of our adult host records are likely in error, or, at a minimum, insecurely documented. On the other hand, there is no doubt that adults do feed on a wide variety of plants. Aside from eggs, *P. spumarius* does not have an inactive or dormant stage. They must feed to live, and feed in quantity. When there are a lot of adults on a particular plant or when adults associate with multiple individuals of the same plant species, chances are they are feeding. This is particularly true of the many adults found on trees and shrubs in dry climate summers, when plants in the herbaceous understory have browned out for the season [22, 44, 45].

**Spittlebug identity in the BRIGIT citizen science data.**   The citizen science survey benefitted from the specific character of the British spittlebug fauna. In many other areas there are common spittlebug species frequenting herbaceous dicots that might easily be misidentified for *P. spumarius* based on observation of spittles alone. This is less the case in Britain, but two British genera, *Neophilaenus* and *Aphrophora*, do include spittlebugs that might be mistaken in the spittle stage for *P. spumarius* if the nymphs themselves are not examined carefully.

The four British *Neophilaenus* species are restricted almost entirely to monocots and can be screened out by eliminating ambiguous records from grasses and sedges. The four *Aphrophora* species are primarily insects of trees and shrubs, but the nymphs of the most common and widespread species, *Aphrophora alni* Fallén, also occasionally feed on herbaceous plants. Thus, although the nymphs are distinct, identification based solely on spittles runs the risk of conflating *P. spumarius* and *A. alni*. In practice, however, extensive field collections across the UK undertaken during the BRIGIT project found very few *A. alni* on herbaceous plants; consequently, this risk is judged to be minimal.

Validation of BRIGIT citizen science host plant records adopted a highly conservative approach: 1) all records based on adult *P. spumarius* were rejected due to the difficulty in distinguishing between sitting and feeding behaviours, as previously described; 2) nymph host records were accepted only where *P. spumarius* identification could be confirmed via accompanying photographs; 3) host records based on the presence of spittle were attributed to *P. spumarius* only when the reported host was an herbaceous dicot.

**Changing plant taxonomy.** Botanical nomenclature poses additional hurdles. Some of our sources are over 100 years old and plant taxonomy has moved on. Wherever possible, we have updated species names to conform with current usage. Where we encountered nomenclatural ambiguity, we used two sources to determine currently valid names: the Integrated Taxonomic Information System [46] and Kew Plants of the World Online [47]. We have updated plant family-level taxonomy to be consistent with contemporary usage, following the template of Christenhusz & Byng [48]. This pared down the number of families by about 9%. In many cases, particularly those involving agricultural or popular sources, we had to make reasonable inferences from common names in multiple languages, an endeavor with its own hazards. We are confident that most of our botanical names are correct and up to date, but in a work of this scale, from such diverse sources, some errors are inevitable. We also recognize the possibility that some of the original plant identifications may have been in error, representing another, hopefully small, source of noise in the data.

## Criteria for inclusion of records in this compilation

We have included all naturally occurring hosts for which we have found at least one usable record. We exclude records based solely on laboratory experiments. Most records are reported to species level. In instances in which we encountered a mix of records that were generic-only and species-specific for a given genus, we include only species-specific records, to avoid the possibility of double counting. This makes our host number estimates more conservative, as, in all likelihood, some of the excluded generic records represent species distinct from but congeneric to those included. Holopainen & Varis [49] report that using this criterion decreased their host total for *Lygus rugulipennis* from 437 to 402, a reduction of 8%. Application of the same criteria to underlying data for the scales *Aspidiotis nerii* and *Hemiberlesia lataninae* [50] reduced the number of species records by about 12.5% in each case. As noted above, we also exclude records of adult hosts based on single specimens.

In cases in which we have multiple records for the same host species, we have given preference in the following order: formal scientific publications (journal articles, books, stand-alone scientific publications), followed by theses and dissertations, followed by more ephemeral Internet sources, followed by unpublished observations by cited observers. Where we have records for multiple geographic areas, we list all areas and cite at least one record for each. For hosts with records for both nymphs and adults, we cite at least one record for each stage. The goal is to keep the list of references for each host species compact, while documenting the

known geographical distribution and observed stages for each host. The 358 citizen science host observations new to science are marked ●●●, while those new to the UK are marked ●●, and those confirming earlier local observations are marked ●.

The geographic units chosen for this report include several with natural boundaries. New Zealand (NZ) and Hawaii (HI) represent discrete island areas where *P. spumarius* has been introduced. The Azores (AZ), where *P. spumarius* is likely but not certainly recently introduced [51–53] are a special island case, but one with very few host records. In North America, where *P. spumarius* is introduced, it has a disjunct distribution [54], divided by the dryer midsection of the continent into discrete eastern (ENA) and western areas (WNA). Britain and Ireland (B&I) represent another natural division. We somewhat arbitrarily divide continental Europe into four areas: Finland and Scandinavia (F&S), Western Europe (WE), Eastern Europe (EE), and the Mediterranean Basin (MED). See the footnotes to Table 1 for the detailed boundaries. A number of Noury's [35] listings are from a source covering "Middle and Northern Europe", a category that does not fit our arbitrary divisions. These records are recorded as M&NE. A few North American references do not specify which section of the continent the records refer to. These are recorded as EorWNA. Although *P. spumarius* is frequent in Kyrgyzstan [55] and is reported across Asia to China and Japan (refs. in [34]), we found only one host record east of European Russia, in Uzbekistan [56]. *P. spumarius* has been reported once, on strawberries, on the island of Réunion in the Indian Ocean [57] but, in the absence of later reports, seems not to have become established.

## Results

Table 1 lists 1311 species of *Philaenus spumarius* host plants, by plant family and binomial in alphabetical order within families. It also gives the life stages observed (nymph and/or adult), the geographical area(s) in which the host association was observed, the BRIGIT citizen science observation status (if any), and selected references.

The headline result is that at 1311 species *P. spumarius* has far more documented host plants than any other herbivorous insect (Table 2). They include ferns, herbs, shrubs, vines and trees, annuals and perennials, grasses and forbs, plants of the tropics, subtropics, temperate and boreal zones, conifers–just about every imaginable kind of vascular plant except those living submerged in aquatic environments. This extraordinary species level diversity is reinforced in the higher order taxonomic diversity, 117 families and 631 genera. Table 3 summarizes the distribution of host species by family for all families represented by 10 or more species.

A large majority of host records, 1113 (84.9%), are for nymphs only. Eighty-eight (6.7%) are for nymphs and adults, and 6.6% (86) for adults alone (in 24 cases the life stage could not be determined or reasonably inferred from the information available). We include about 1890 geographical area records. Most hosts have been recorded from a single geographical area, but many are recorded from two or more. Multiple area hosts are typically widespread weedy herbs or common garden plants. Europe and North America account for most geographical records, 73.2% and 19.6% respectively, but Hawaii (4.1%) and New Zealand (2.8%) are well represented for their size. Britain and Ireland loom especially large, alone accounting for 37.2% of all records, in major part a reflection of the BRIGIT program. BRIGIT citizen science records include 358 hosts that are not duplicated in preexisting sources, a full 27.3% of all recorded hosts. Another 198 (15.1%) represent records that are new to Britain and Ireland. Seventy-two (5.9%) represent confirmation of hosts recorded for Britain and Ireland in preexisting sources. In total, BRIGIT citizen science records include 628 (47.9%) of the 1311 recorded *P. spumarius* hosts.

**Table 1. *Philaenus spumarius* host plants, by plant family, life stage, geographic occurrence, and BRIGIT project status.** See footnotes for a key to and explanation of abbreviations.

| Host species by family | Stage | Geographic area(s) | Cit. sci. obs. | Selected record source(s) |
|---|---|---|---|---|
| **Acanthaceae** | | | | |
| *Acanthus mollis* | N | B&I | ••• | |
| **Adoxaceae** | | | | |
| *Sambucus canadensis* | N | ENA | | [58] |
| *Sambucus nigra* | N | B&I WE WNA | •• | [25, 26, 29] |
| *Sambucus racemosa* | N | F&S WE | | [35, 36] |
| **Aizoaceae** | | | | |
| *Carpobrotus chilensis* | N | B&I | ••• | |
| *Carpobrotus edulis* | N,A | B&I MED WNA | •• | [59, 60] |
| *Cleretum bellidiforme* | N | B&I | ••• | |
| *Sesuvium portulacastrum* | N | HI | | [27] |
| **Alismataceae** | | | | |
| *Alisma triviale* | N | ENA | | [58] |
| **Alstroemeriaceae** | | | | |
| *Alstroemeria* sp. | N | B&I | | HMBL-England |
| **Altingiaceae** | | | | |
| *Liquidambar styraciflua* | N | B&I | ••• | |
| **Amaranthaceae** | | | | |
| *Alternanthera dentata* | N | B&I | ••• | |
| *Amaranthus retroflexus* | A | EE | | [61] |
| *Atriplex prostrata* | A | WNA | | VT-CA-2013-A |
| *Beta vulgaris* | N | B&I MED ENA HI | •• | [5, 27, 62] |
| *Chenopodium album* | N,A | F&S B&I WE MED ENA | • | [5, 35, 36, 63, 64] |
| *Spinacia oleracea* | N | B&I M&NE EorWNA | •• | [35, 65] |
| **Amaryllidaceae** | | | | |
| *Allium cepa* | N | B&I ENA | •• | [5] |
| *Allium sativum* | N | B&I | ••• | |
| *Allium schoenoprasum* | N | F&S B&I | •• | [36] |
| *Allium schubertii* | N | B&I | ••• | |
| *Allium siculum* | N | B&I | ••• | |
| *Allium sphaerocephalon* | N | B&I | ••• | |
| *Allium triquetrum* | N | B&I | ••• | |
| *Allium vineale* | N | ENA | | [32] |
| *Allium × proliferum* | N | B&I | | J. Owen |
| *Narcissus* sp. | A | ENA | | [66] |
| **Anacardiaceae** | | | | |
| *Cotinus coggygria* | N | B&I WE | •• | [67] |
| *Pistacia lentiscus* | N,A | MED | | [68, 69] |
| *Pistacia terebinthus* | A | MED | | [68] |
| *Rhus glabra* | N | ENA | | [32] |
| *Toxicodendron diversilobum* | N | WNA | | [29] |
| *Toxicodendron radicans* | N | ENA | | [32] |
| **Apiaceae** | | | | |
| *Aciphylla* sp. | ? | NZ | | [70] |
| *Aegopodium podagraria* | N | F&S B&I WE | •• | [25, 26, 36] |
| *Ammi majus* | N | B&I | ••• | |

*(Continued)*

**Table 1.** (Continued)

| Host species by family | Stage | Geographic area(s) | Cit. sci. obs. | Selected record source(s) |
|---|---|---|---|---|
| *Ammi visnaga* | N | MED | | [71] |
| *Anethum graveolens* | N | B&I | ●●● | |
| *Angelica archangelica* | N | F&S | | [36] |
| *Angelica sylvestris* | N | F&S B&I | ●● | [36] |
| *Anthriscus cerefolium* | N | B&I | | J. Owen |
| *Anthriscus sylvestris* | N | F&S B&I WE MED | ● | [9, 36, 63, 72] |
| *Anthriscus vulgaris* | N | WE | | [73] |
| *Apium graveolens* | N | B&I MED ENA WNA HI | ●● | [5, 29, 42, 74] |
| *Apium nodiflorum* | N | B&I | ●●● | |
| *Astrantia major* | N | B&I | ●●● | |
| *Carum carvi* | N | WE | | [26] |
| *Caucalis platycarpos* | N | MED | | [75] |
| *Cervaria* sp. | N | MED | | [10] |
| *Chaerophyllum bulbosum* | N | M&NE | | [35] |
| *Chaerophyllum hirsuitum* | N | M&NE | | [35] |
| *Chaerophyllum temulum* | N | WE | | [35] |
| *Conium maculatum* | N | WNA | | [29] |
| *Conopodium majus* | N | B&I | | [15, 63] |
| *Coriandrum sativum* | N | B&I | ●●● | |
| *Crithmum maritimum* | N | B&I | ●●● | |
| *Daucus carota* | N,A | F&S B&I MED ENA WNA HI NZ | ●● | [5, 7, 28, 29, 36, 37, 76] |
| *Eryngium campestre* | N | MED | | [77] |
| *Eryngium maritimum* | N | B&I | ●●● | |
| *Eryngium planum* | N | B&I | ●●● | |
| *Falcaria vulgaris* | N | WE | | [25, 26] |
| *Foeniculum vulgare* | N | B&I MED WNA | ●● | [9, 10, 60, 78] VT-CA-2021-N |
| *Heracleum mantegazzianum* | N | B&I | ●●● | |
| *Heracleum maximum* | N,A | WNA | | [79] VT-WA-2003-A |
| *Heracleum sphondylium* | N | B&I WE | ● | [25, 26, 37] |
| *Laserpitium latifolium* | N | M&NE | | [35] |
| *Levisticum officinale* | N | B&I WE ENA | ●● | [35, 80] |
| *Ligusticum scoticum* | N,A | B&I | | [81] |
| *Myrrhis odorata* | N | B&I | ●●● | |
| *Pastinaca sativa* | N | B&I WE ENA HI | ● | [5, 25, 26, 28, 63] |
| *Petroselinum crispum* | N | B&I WE ENA WNA HI | ● | [5, 29, 37, 42, 82] |
| *Peucedanum alsaticum* | N | M&NE | | [35] |
| *Peucedanum officinale* | N | B&I | | HMBL-England |
| *Peucedanum palustre* | N | F&S | | [36] |
| *Pimpinella anisum* | A | MED | | [74] |
| *Pimpinella major* | N | WE | | [25] |
| *Pimpinella saxifraga* | N,A | F&S B&I WE MED | ●● | [35, 36, 74] |
| *Sanicula liberta* | N | WNA | | [29] |
| *Scandix pecten-veneris* | N | MED | | [6] |
| *Seseli tortuosum* | N | MED | | [60] |
| *Smyrnium olusatrum* | N | B&I | ●●● | |
| *Tordylium* sp. | N | MED | | [10] |
| *Torilis japonica* | N | WE | ●● | [25] |

*(Continued)*

**Table 1.** (Continued)

| Host species by family | Stage | Geographic area(s) | Cit. sci. obs. | Selected record source(s) |
|---|---|---|---|---|
| *Torilis nodosa* | N | MED | | [8, 60] |
| **Apocynaceae** | | | | |
| *Alyxia stellata* | N | HI | | [27] |
| *Asclepias* sp. | N | ENA | | [32] |
| *Trachelospermum jasminoides* | N | B&I | ••• | |
| *Vinca major* | N | B&I WNA | •• | [29] |
| *Vinca minor* | N | B&I | ••• | |
| **Aquifoliaceae** | | | | |
| *Ilex anomala* | N | HI | | [28] |
| *Ilex verticillata* | N | B&I | ••• | |
| **Araceae** | | | | |
| *Acorus calamus* | N | B&I | ••• | |
| *Arum italicum* | N | WE | | [83] |
| *Zantedeschia aethiopica* | N | B&I WNA | •• | [29] |
| **Araliaceae** | | | | |
| *Hedera canariensis* | N | WNA | | [29] |
| *Hedera colchica* | N | B&I | ••• | |
| *Hedera helix* | N,A | B&I WNA | • | [29, 37, 84] |
| **Asparagaceae** | | | | |
| *Aloe vera* | N | B&I | ••• | |
| *Asparagus officinalis* | A | MED | | [85] |
| *Chlorogalum pomeridianum* | N | WNA | | [29] |
| *Chlorophytum laxum* | N | B&I | ••• | |
| *Convallaria majalis* | N | F&S | | [36] |
| *Cordyline australis* | N | B&I | ••• | |
| *Cordyline fruticosa* | N | HI | | [28] |
| *Eucomis comosa* | N | B&I | ••• | |
| *Hyacinthoides hispanica* | N | B&I | ••• | |
| *Hyacinthoides non-scripta* | N | B&I | • | [37] |
| *Maianthemum bifolium* | N | F&S | | [36] |
| *Muscari comosum* | N | MED | | [11] |
| *Ornithogalum ortophyllum* | N | MED | | [86] |
| *Ruscus aculeatus* | N | MED | | [86] |
| **Asphodelaceae** | | | | |
| *Asphodelus ramosus* | N | MED | | [44] |
| *Hemerocallis fulva* | N | ENA | | [5] |
| *Hemerocallis lilioasphodelus* | N | B&I | ••• | |
| *Kniphofia galpinii* | N | B&I | ••• | |
| *Simethis planifolia* | N | MED | | [60] |
| **Aspleniaceae** | | | | |
| *Athyrium filix-femina* | N | F&S B&I | •• | [36] |
| *Thelypteris palustris* | N | ENA | | [87] |
| **Asteraceae** | | | | |
| *Acanthospermum australe* | N | HI | | [27] |
| *Achillea ageratum* | N | B&I | ••• | |
| *Achillea filipendulina* | N | B&I | ••• | J. Owen |
| *Achillea millefolium* | N | F&S B&I WE ENA WNA NZ | • | [36, 37, 63, 87, 88] VT-CA-2021-N |

(Continued)

**Table 1.** (Continued)

| Host species by family | Stage | Geographic area(s) | Cit. sci. obs. | Selected record source(s) |
|---|---|---|---|---|
| *Achillea ptarmica* | N | F&S | ●● | [36] |
| *Adenostyles alpina* | N | MED | | HMBL-Italy |
| *Ambrosia artemisiifolia* | N,A | EE ENA | | [32, 89] |
| *Ambrosia trifida* | N | ENA | | [32] |
| *Anaphalis margaritacea* | N,A | ENA WNA | | [87, 90, 91] |
| *Andryala integrifolia* | N | WE MED | | [25, 92] |
| *Antennaria plantaginifolia* | N | ENA | | [5] |
| *Anthemis arvensis* | N | WE | | [25] |
| *Anthemis arvensis* | N | MED | | [7, 11, 69] |
| *Anthemis chia* | N | MED | | [6] |
| *Anthemis cotula* | N | MED | | [93] |
| *Arctium lappa* | N | WE | | [25] |
| *Arctium minus* | ? | ENA | | [94] |
| *Arctium tormentosum* | N | F&S | | [36] |
| *Arctotheca calendula* | ? | MED | | [95] |
| *Argyranthemum frutescens* | N | B&I WE | | J. Owen [35] |
| *Arnica montana* | N | B&I | ●●● | |
| *Artemisia abrotanum* | N | B&I WE | ●● | [35] |
| *Artemisia absinthium* | N | F&S | | [36] |
| *Artemisia campestris* | N | WE, MED | | [25, 60] |
| *Artemisia caudata* | N | ENA | | VT-MN-2001-N |
| *Artemisia dracunculus* | N | B&I ENA | ●● | [5] |
| *Artemisia tridentata* | A | WNA | | VT-UT-2003-A |
| *Artemisia verlotiorum* | A | WE | | [96] |
| *Artemisia vulgaris* | N | F&S B&I WE MED WNA HI | ●● | [9, 28, 29, 36, 72] |
| *Aster alpinus* | N | B&I | ●●● | |
| *Aster amellus* | N | B&I | ●●● | |
| *Asteriscus* sp. | ? | MED | | [97] |
| *Baccharis pilularis* | N | WNA | | [29] |
| *Bahia ambrosioides* | N | B&I | ●●● | |
| *Bellis perennis* | N | B&I MED ENA | ●● | [33, 98] |
| *Bidens pilosa* | N | HI | | [28] |
| *Bidens tripartida* | N | F&S WE | | [35, 36] |
| *Brachyglottis grayi* | N | B&I | ●●● | |
| *Buphthalmum salicifolium* | N | B&I M&NE | ●● | [35] |
| *Calendula arvensis* | N | MED | | [7, 11] |
| *Calendula officinalis* | N | B&I MED | ●● | [75] |
| *Carduus acanthoides* | N | NZ | | [37] |
| *Carduus crispus* | N | B&I | ●●● | |
| *Carduus nutans* | N,A | B&I ENA | | [63, 99] |
| *Carduus pycnocephalus* | N,A | WNA | | [100] |
| *Carduus tenuiflorus* | N | MED | | [77] |
| *Carlina hispanica* | A | MED | | [7] |
| *Catananche caerulea* | N | B&I | ●●● | |
| *Celmisia* sp. | ? | NZ | | [70] |
| *Centaurea aspera* | N | B&I | | HMBL-England |
| *Centaurea centaurium* | N | B&I | ●●● | |

(*Continued*)

**Table 1.** (Continued)

| Host species by family | Stage | Geographic area(s) | Cit. sci. obs. | Selected record source(s) |
|---|---|---|---|---|
| *Centaurea cyanus* | N | B&I, WE | ●● | [25] |
| *Centaurea jacea* | N | M&NE | | [35] |
| *Centaurea macrocephala* | N | B&I | ●●● | |
| *Centaurea montana* | N | B&I | ●●● | |
| *Centaurea nigra* | N | B&I | ● | [37] |
| *Centaurea ornata* | A | MED | | [7] |
| *Centaurea phrygia* | N | F&S | | [36] |
| *Centaurea scabiosa* | N | B&I | ●●● | |
| *Centaurea solstitialis* | N,A | WNA | | [101] |
| *Chondrilla juncea* | N | WE MED | | [6, 25, 92] |
| *Chrysanthemum carinatum* | N | F&S | | [36] |
| *Chrysanthemum makinoi* | N | WE | | [35] |
| *Chrysanthemum maximum* | N | WNA HI | | [28, 29] |
| *Chrysanthemum vulgare* | N | WE | | [25] |
| *Chrysanthemum x morifolium* | N | B&I | ●●● | |
| *Cicerbita alpina* | N | B&I | ●●● | |
| *Cichorium intybus* | N | MED ENA | ●● | [11, 32, 75] |
| *Cirsium arvense* | N,A | F&S B&I WE MED ENA NZ | ● | [7, 36, 37, 84, 87, 88] |
| *Cirsium discolor* | N | ENA | | [102] |
| *Cirsium dissectum* | N | B&I | ●●● | |
| *Cirsium eriophorum* | N | B&I | | [63] |
| *Cirsium heterophyllum* | N | F&S | | [36] |
| *Cirsium lanceolatum* | N | WNA | | [29] |
| *Cirsium oleraceum* | N | M&NE | | [35] |
| *Cirsium palustre* | N | F&S B&I WNA | ●● | [36, 103] |
| *Cirsium rivulare* | N | B&I | ●●● | |
| *Cirsium vulgare* | N | F&S B&I ENA WNA | ●● | [5, 36, 103, 104] |
| *Coleostephus myconis* | N | MED | | [92] |
| *Conyza sumatrensis* | N | HI | | [28] |
| *Coreopsis* sp. | ? | ? | | [57] |
| *Cosmos atrosanguineus* | N | B&I | ●●● | |
| *Crepis biennis* | N | B&I WE MED | ●● | [11, 88] |
| *Crepis capillaris* | N | B&I WE NZ | ●● | [26, 37] |
| *Crepis neglecta* | N | MED | | [11] |
| *Crepis paludosa* | N | F&S | | [36] |
| *Crepis vesicaria* | N | B&I | ●●● | |
| *Cynara cardunculus* | N | B&I WE MED WNA | ●● | [29, 35, 105] |
| *Cynara scolymus* | N | B&I | ●●● | |
| *Dahlia* sp. | N | WE WNA HI | | [25, 28, 29] |
| *Dimorphotheca pluvialis* | N | B&I | ●●● | |
| *Dittrichia viscosa* | N,A | MED | | [7, 11, 106] |
| *Dubautia scabra* | N | HI | | [28] |
| *Echinacea pallida* | N | B&I | ●●● | |
| *Echinops bannaticus* | N | B&I | ●●● | |
| *Echinops ritro* | N | B&I | ●●● | |
| *Echinops sphaerocephalus* | N | B&I | ●●● | |
| *Erigeron annuus* | N | ENA NZ | | [37, 107] |

*(Continued)*

**Table 1.** (Continued)

| Host species by family | Stage | Geographic area(s) | Cit. sci. obs. | Selected record source(s) |
|---|---|---|---|---|
| *Erigeron bonariensis* | N | NZ | | [37] |
| *Erigeron canadensis* | N | WE MED ENA | | [5, 25, 60] |
| *Erigeron glaucus* | N | WNA | | [108] |
| *Erigeron karvinskianus* | N | B&I | ●●● | |
| *Erigeron philadelphicus* | A | WNA? | | UCRC |
| *Erigeron strigosus* | N | ENA | | [32] |
| *Erigeron sumatrensis* | N | B&I HI | ●● | [28] |
| *Eupatorium cannabinum* | N | B&I | | CH-England-2019-N |
| *Eurybia divaricata* | N | B&I | ●●● | |
| *Euthenia graminifolia* | N | ENA | | VT-MN-2001-N |
| *Eutrochium maculatum* | N | B&I | ●●● | |
| *Felicia petiolata* | N | WNA | | [29] |
| *Galactites tomentosa* | N | MED | | [8] |
| *Galinsoga ciliata* | A | ENA | | [109] |
| *Galinsoga parviflora* | A | ENA | | [109] |
| *Gamochaeta purpurea* | N | HI | | [28] |
| *Gazania rigens* | N | WNA | | [29] |
| *Glebionis coronaria* | N | F&S MED | | [11, 36] |
| *Gnaphalium sylvaticum* | N | M&NE | | [35] |
| *Hedypnois cretica* | N | MED | | [11] |
| *Helianthus annuus* | N | B&I | ●●● | |
| *Helianthus tuberosus* | N | B&I | ●●● | |
| *Helicanthus giganteus* | N | B&I | ●●● | |
| *Helichrysum italicum* | N | B&I MED | ●● | [60] |
| *Heliopsis scabra* | N | WE | | [35] |
| *Helminthoteca echioides* | N | B&I MED WNA | | [6, 7, 9, 110] |
| *Hemizonia congesta* | ? | WNA | | [111] |
| *Hieracium caespitosum* | N,A | F&S ENA NZ | | [36, 70, 87, 112] |
| *Hieracium lepidulum* | N,A | NZ | | [70] |
| *Hieracium maculatum* | N | B&I | ●●● | |
| *Hieracium murorum* | N | MED | | HMBL-Italy |
| *Hieracium pilosella* | N,A | NZ | | [70] |
| *Hieracium praealtum* | N | NZ | | [112] |
| *Hieracium tridentatum* | N | WE | | [25] |
| *Hieracium umbellatum* | N | F&S | | [36] |
| *Hyoseris* sp. | N | MED | | [10] |
| *Hypochaeris glabra* | N | MED,WNA | | [29, 92] |
| *Hypochaeris radicata* | N,A | B&I MED ENA WNA HI | ● | [5, 28, 37, 40] SEMC |
| *Hypochoeris maculata* | N | F&S | | [36] |
| *Inula* sp. | N | MED | | [10] |
| *Jacobaea aquatica* | N | B&I | ●●● | |
| *Jacobaea maritima* | N | B&I | ●●● | |
| *Jacobaea paludosa* | N | B&I | ●●● | |
| *Jacobaea vulgaris* | N | B&I | ●●● | |
| *Lactuca muralis* | N | B&I | ●●● | |
| *Lactuca sativa* | N | B&I EorWNA HI | ●● | [28, 65] |
| *Lactuca serriola* | N | B&I ENA | ●● | [104] |

(*Continued*)

**Table 1.** (Continued)

| Host species by family | Stage | Geographic area(s) | Cit. sci. obs. | Selected record source(s) |
|---|---|---|---|---|
| *Lactuca virosa* | N | ENA | | [5] |
| *Lapsana communis* | N | B&I WE | ●● | [25, 26] |
| *Leontodon hispidus* | N | B&I WE | ●● | [26] |
| *Leontodon taraxacoides* | N | NZ | | [37] |
| *Leucanthemum merinoi* | N | MED | | [60] |
| *Leucanthemum vulgare* | N | B&I WE ENA | ●● | [32, 35] |
| *Leucanthemum x superbum* | N | B&I | ●●● | |
| *Leucophyta brownii* | N | B&I | ●●● | |
| *Logfia gallica* | N | MED | | [92] |
| *Madia elegans* | N | WNA | | [29] |
| *Matricaria camomilla* | N | WE | | [82] |
| *Matricaria discoidea* | N | WE | | [35] |
| *Oclemena acuminata* | N | ENA | | [87] |
| *Oclemena nemoralis* | N | ENA | | [87] |
| *Onopordum acanthium* | A | MED | | [7] |
| *Ozothamnus rosmarinifolius* | N | B&I | | HMBL-England |
| *Pallenis spinosa* | N | MED | | [11] |
| *Petasites hybridus* | N | B&I WE | ●● | [25] |
| *Petasites pyrenaicus* | N | B&I WE | ●● | [35] |
| *Picris hieracioides* | N | MED | | [11] |
| *Pilosella aurantiaca* | N | B&I ENA | ●● | [87] |
| *Pilosella caespitosa* | N,A | B&I ENA NZ | ●● | [33, 70] |
| *Pilosella officinarum* | N | B&I | ●●● | |
| *Pluchea odorata* | N | HI | | [28] |
| *Psephellus dealbatus* | N | B&I | ●●● | |
| *Pulicaria dysenterica* | N | B&I | ●●● | |
| *Reichardia* sp. | N | MED | | [10, 78] |
| *Rhaponticum carthamoides* | N,A | WE | | [113] |
| *Rhaponticum scariosum* | N | B&I | ●●● | |
| *Rudbeckia fulgida* | N | B&I | ●●● | |
| *Rudbeckia hirta* | N | F&S B&I | ●● | [114] |
| *Rudbeckia laciniata* | N | WNA | ●● | [29] |
| *Santolina chamaecyparissus* | N | B&I | ●● | [115] |
| *Scolymus hispanicus* | N | MED | | [77, 116] |
| *Scorzonera aristata* | N | MED | | [11] |
| *Scorzoneroides autumnalis* | N | F&S B&I WE | ●● | [26, 36] |
| *Senecio jacobaea* | N | B&I | | [37, 63] |
| *Senecio mikanioides* | N | HI | | [28] |
| *Senecio squalidus* | N | B&I | ●●● | |
| *Senecio vulgaris* | N | MED | | [60, 69] |
| *Silybum marianum* | N | B&I MED | ●● | [6] |
| *Solidago altissima* | N,A | WE ENA HI | | [28, 102, 117, 118] |
| *Solidago canadensis* | N | F&S B&I | ●● | [36] |
| *Solidago gigantea* | N | WE ENA | | VT-MN-2001-N |
| *Solidago rugosa* | N | ENA | | [87] |
| *Solidago sempervirens* | N | ENA | | [87] |
| *Solidago shortii* | A | ENA | | [119] |

(*Continued*)

**Table 1.** (Continued)

| Host species by family | Stage | Geographic area(s) | Cit. sci. obs. | Selected record source(s) |
|---|---|---|---|---|
| *Solidago virgaurea* | N | F&S B&I WE | ●● | [35, 36] |
| *Sonchus arvensis* | N | B&I MED | ●● | [11] |
| *Sonchus asper* | N | B&I MED WNA | ●● | [9, 29] |
| *Sonchus oleraceus* | N | B&I MED WNA HI NZ | ●● | [6, 7, 9, 28, 29, 37, 69] |
| *Sonchus terrenimus* | N | MED | | [92] |
| *Stokesia laevis* | N | B&I | ●●● | |
| *Symphyotrichum cordifolium* | N | B&I ENA | ●● | [104] |
| *Symphyotrichum ericoides* | N | ENA | | [104] |
| *Symphyotrichum lanceolatus* | N | M&NE | | [35] |
| *Symphyotrichum lateriflorum* | N | B&I | ●●● | |
| *Symphyotrichum novae-angliae* | N | B&I | ●●● | |
| *Symphyotrichum novi-belgii* | N | B&I | ●●● | |
| *Symphyotrichum tradescantii* | N | B&I | | [120] |
| *Symphyotrichum turbinellum* | N | B&I | ●●● | |
| *Symphytum officinale* | N | B&I WE | ●● | [82] |
| *Tagetes lucida* | N | B&I | ●●● | |
| *Tanacetum parthenium* | N | B&I | ●●● | |
| *Tanacetum vulgare* | N | F&S B&I WE WNA | ●● | [29, 35, 36] |
| *Taraxacum kok-saghyz* | N | B&I | ●●● | |
| *Taraxacum officinale* | N | B&I WE, MED ENA WNA NZ | ● | [5, 25, 29, 37, 40, 75, 83] |
| *Tolpis umbellata* | N | MED | | [11, 69] |
| *Tragopogon dubius* | N | B&I | ●●● | |
| *Tragopogon porrifolius* | N | WNA | | [29] |
| *Tragopogon pratensis* | N | B&I | ●●● | |
| *Tripleurospermum inodorum* | N | F&S B&I WE | ●● | [25, 36] |
| *Tripolium pannoncium* | N | B&I | ●●● | |
| *Tussilago farfara* | N | F&S B&I | ●● | [36] |
| *Urospermum dalechampii* | N | MED | | [9, 11, 69] |
| *Urospermum picroides* | N | MED | | [11] |
| *Wyethia amplexicaulis* | A | WNA | | AMNH |
| **Balsaminaceae** | | | | |
| *Impatiens capensis* | N | ENA | | [87] |
| *Impatiens glandulifera* | N | B&I | ●●● | |
| *Impatiens noli-tangere* | N | WE | | [26] |
| **Berberidaceae** | | | | |
| *Berberis darwinii* | N | B&I | ● | [39] |
| *Berberis julianae* | N | B&I | ●●● | |
| *Berberis thunbergii* | N | B&I ENA | ●● | [5] |
| *Mahonia aquifolium* | N | B&I | ● | [39] |
| *Nandina domestica* | N | B&I | ●●● | |
| *Podophyllum peltatum* | N | B&I | ●●● | |
| **Betulaceae** | | | | |
| *Alnus glutinosa* | N | F&S B&I | ●● | [36, 121] |
| *Alnus incana* | N | F&S | | [36] |
| *Alnus rubra* | A | WNA | | VT-OR-2003-A |
| *Betula alleghaniensis* | A | ENA | | [122] |
| *Betula nigra* | N | B&I | ●●● | |

(*Continued*)

**Table 1.** (Continued)

| Host species by family | Stage | Geographic area(s) | Cit. sci. obs. | Selected record source(s) |
|---|---|---|---|---|
| *Betula papyrifera* | A | ENA | | [122] |
| *Betula pendula* | N | F&S | | [36] |
| *Betula populifolia* | A | ENA | | [122] |
| *Betula pubescens* | N | F&S | ●● | [36, 121] |
| *Carpinus orientalis* | ? | MED | | [97] |
| *Corylus avellana* | N,A | F&S B&I WE | ●● | [67, 121] |
| *Ostrya carpinifolia* | A | MED | | [123] |
| **Boraginaceae** | | | | |
| *Amsinckia menziesii* | N | WNA | | RKpercom |
| *Amsinckia spectabilis* | N | WNA | | RKpercom |
| *Anchusa officinalis* | N | WE | | [26] |
| *Asperugo procumbens* | N | WE | | [25] |
| *Borago officinalis* | N | B&I MED | ●● | [9, 116] |
| *Echium pininana* | N | B&I | ●●● | |
| *Echium plantagineum* | N | MED | | [7, 92] |
| *Echium vulgare* | N | F&S, B&I MED | ●● | [75, 124] |
| *Glandora diffusa* | N | B&I | ●●● | |
| *Heliotropium arborescens* | N | WNA | | [29] |
| *Lithospermum croceum* | N | ENA | | VT-MN-2001-N |
| *Lithospermum officinale* | N | B&I | | [63] |
| *Myosotis arvensis* | N | B&I | ●●● | |
| *Myosotis azorica* | N | HI | | [28] |
| *Myosotis scorpioides* | N | B&I WNA | ●● | [29] |
| *Pentaglottis sempervirens* | N | B&I | ●●● | |
| *Phacelia tanacetifolia* | N | B&I | ●●● | |
| *Symphytum x uplandicum* | N | B&I | ●●● | |
| **Brassicaceae** | | | | |
| *Alliaria petiolata* | N | B&I M&NE | ●● | [35, 125] |
| *Alyssum* sp. | A | EorWNA | | [126] |
| *Arabis caucasica* | N | B&I | | J. Owen |
| *Armoracia rusticana* | N | B&I WE | ●● | [25] |
| *Aubrieta deltoidea* | N | B&I | | J. Owen |
| *Barbarea stricta* | N | WE | | [25] |
| *Barbarea verna* | N | B&I ENA | ●● | [32] |
| *Barbarea vulgaris* | N | ENA | | [5] |
| *Brassica napus* | N | B&I WE | ●● | [25, 26] |
| *Brassica nigra* | ? | ENA | | [33] |
| *Brassica oleracea* | N,A | B&I ENA HI | ● | [28, 32, 39] |
| *Brassica rapa* | N | B&I | ●●● | |
| *Calepina* sp. | N | MED | | [10, 40] |
| *Capsella bursa-pastoris* | N | B&I WE MED ENA NZ | ●● | [5, 9, 25, 26, 37] |
| *Cardamine flexuosa* | N | B&I | | JRpercom |
| *Cardamine pratensis* | N | B&I WE | ● | [72, 115] Nickel |
| *Descurainia sophia* | N | WE | | [25] |
| *Diplotaxis tenuifolia* | N | B&I | ●●● | |
| *Eruca vesicaria* | N | B&I | ●●● | |
| *Erysimum cheiranthoides* | N | WE WNA | | [25, 26, 29] |

*(Continued)*

**Table 1.** (Continued)

| Host species by family | Stage | Geographic area(s) | Cit. sci. obs. | Selected record source(s) |
|---|---|---|---|---|
| *Erysimum cheiri* | N | B&I M&NE | | J. Owen [35] |
| *Hesperis matronalis* | N | B&I WE | ●● | [25] |
| *Iberis sempervirens* | A | B&I | | J. Owen |
| *Lepidium campestre* | N | WE ENA | | [5, 25] |
| *Lepidium didymum* | N | NZ | | [37] |
| *Lepidium draba* | N | B&I, MED | | [11] HMBL-England |
| *Lepidium ruderale* | N | WE | | [25, 26] |
| *Lepidium virginicum* | N | ENA | | [5] |
| *Lunaria annua* | N | B&I | ●●● | |
| *Matthiola incana* | N | WNA | | [29] |
| *Myagrum perfoliatum* | N | MED | | [8] |
| *Nasturtium officinale* | N | B&I | ●●● | |
| *Raphanus raphanistrum* | N | B&I WNA HI | ●● | [29, 127] |
| *Rapistrum* sp. | N | MED | | [40] |
| *Rorippa amphibia* | N | WE | | [35] |
| *Sinapis arvensis* | N | B&I | | HMBL-England |
| *Sisymbrium altissimum* | N | WE, ENA | | [5, 25] |
| *Sisymbrium officinale* | N | B&I | ●●● | |
| *Thlaspi arvense* | N | F&S ENA | | [33, 36] |
| **Bromeliaceae** | | | | |
| *Fascicularia bicolor* | N | B&I | ●●● | |
| **Buxaceae** | | | | |
| *Buxus sempervirens* | N | B&I | ●●● | |
| *Sarcococca confusa* | N | B&I | ●●● | |
| **Campanulaceae** | | | | |
| *Campanula patula* | N | WE | | [25] |
| *Campanula persicifolia* | N | WNA | | [29] |
| *Campanula glomerata* | N | B&I M&NE | ●● | [35] |
| *Campanula latifolia* | N | B&I | ●●● | |
| *Campanula medium* | N | B&I ENA | ●● | [5] |
| *Campanula patula* | N | F&S | | [36] |
| *Campanula persicifolia* | N | F&S B&I | ●● | [36] |
| *Campanula portenschlagiana* | N | B&I | ●●● | |
| *Campanula poscharskyana* | N | B&I | ●●● | |
| *Campanula punctata* | N | B&I | ●●● | |
| *Campanula pyramidalis* | N | B&I | ●●● | |
| *Campanula rapunculus* | N | B&I | ●●● | |
| *Campanula rotundifolia* | N | F&S B&I | ●● | [36] |
| *Campanula trachelium* | N | F&S B&I | | [36] J. Owen |
| *Jasione maritima* | N | B&I MED | ●● | [60] |
| *Legousia* sp. | N | MED | | [8] |
| *Lobelia cardinalis* | N | B&I | ●●● | |
| *Lobelia gibbosa* | N | B&I | ●●● | |
| *Phyteuma nigrum* | N | MED | | HMBL-Italy |
| **Cannabaceae** | | | | |
| *Cannabis sativa* | N | MED, ENA | | [128] |
| *Humulus lupulus* | N | WE | | [25] |

(*Continued*)

**Table 1.** (Continued)

| Host species by family | Stage | Geographic area(s) | Cit. sci. obs. | Selected record source(s) |
|---|---|---|---|---|
| **Caprifoliaceae** | | | | |
| *Centranthus calcitrapae* | N | MED | | [60] |
| *Centranthus ruber* | N | B&I WNA | ●● | [29] |
| *Cephalaria gigantea* | N | B&I | ●●● | |
| *Diervilla* sp. | A | EorWNA | | [126] |
| *Dipsacus fullonum* | N | B&I WE MED ENA | ● | [5, 25, 63, 98] |
| *Knautia arvensis* | N | F&S B&I WE | ● | [26, 36, 63] |
| *Knautia macedonica* | N | B&I | ●●● | |
| *Knautia sylvatica* | N | WE | | [88] |
| *Linnaea amabilis* | N | B&I | ●●● | |
| *Linnaea x grandiflora* | N | B&I | ●●● | |
| *Lomelosia* sp. | A | MED | | [7] |
| *Lonicera caerulea* | N | B&I | ●●● | |
| *Lonicera caprifolium* | N | B&I | ●●● | |
| *Lonicera etrusca* | N | B&I | ●●● | |
| *Lonicera japonica* | N,A | B&I HI NZ | ●● | [27, 129] |
| *Lonicera nitida* | N | B&I | ●●● | |
| *Lonicera periclymenum* | N | B&I | ●●● | |
| *Lonicera pileata* | N | B&I | ●●● | |
| *Scabiosa atropurpurea* | N | MED | | [11] |
| *Scabiosa columbaria* | N | B&I | ●●● | |
| *Succisa pratensis* | N | F&S B&I | ●● | [36] |
| *Succisella inflexa* | N | B&I | ●●● | |
| *Symphoricarpos albus* | N | B&I | ●●● | |
| *Valeriana excelsa* | N | F&S | | [36] |
| *Valeriana officinalis* | N | B&I WE | ●● | [35] |
| *Valeriana sambucifolia* | N | F&S | | [36] |
| *Valerianella locusta* | N | B&I | ●●● | |
| *Viburnum lantana* | N | M&NE | | [35] |
| *Viburnum opulus* | N | B&I | ●●● | |
| *Viburnum sargentii* | N | B&I | ●●● | |
| *Viburnum tinus* | N | B&I | ●●● | |
| *Viburnum x bodnantense* | N | B&I | ●●● | |
| *Viburnum x carlcephalum* | N | B&I | ●●● | |
| *Weigela* sp. | N | WE ENA | | [32, 43] |
| **Caryophyllaceae** | | | | |
| *Agrostemma githago* | N,A | B&I WE ENA | ●● | [25, 32] |
| *Cerastium arvense* | N | WE NZ | | [25, 37] |
| *Cerastium brachypetalum* | N | MED | | [92] |
| *Cerastium fontanum* | N | B&I ENA | ●● | [32] |
| *Cerastium glomeratum* | N | M&NE MED NZ | | [6, 35, 37, 40] |
| *Cerastium holosteoides* | N | B&I ENA NZ | | [5, 37] |
| *Dianthus armeria* | N | B&I | ●●● | |
| *Dianthus barbatus* | N | B&I WNA | ●● | [29] |
| *Dianthus carthusianorum* | N | B&I | ●●● | |
| *Dianthus caryophyllus* | N | B&I WNA | ●● | [29] |
| *Dianthus chinensis* | N | HI | | [28] |

*(Continued)*

**Table 1.** (Continued)

| Host species by family | Stage | Geographic area(s) | Cit. sci. obs. | Selected record source(s) |
|---|---|---|---|---|
| *Dianthus gratianopolitanus* | N | B&I | ●●● | |
| *Dianthus plumarius* | N | B&I | ●●● | |
| *Lychnis* sp. | ? | ? | | [57] |
| *Moehringia trinervia* | N | B&I WE | ●● | [35] |
| *Rabelera holostea* | N | F&S B&I WE | ● | [35–37] |
| *Sagina apetala* | N | WE | | [35] |
| *Saponaria officinalis* | N | F&S B&I WE ENA WNA | ●● | [29, 32, 35, 36] |
| *Scleranthus annuus* | N | WE | | [26] |
| *Silene coronaria* | N | B&I ENA | ●● | [33] |
| *Silene dioica* | N | F&S B&I | ●● | [36] |
| *Silene flos-cuculi* | N | F&S B&I WE | ● | [36, 72, 115] |
| *Silene gallica* | N | MED | | [92] |
| *Silene latifolia* | N | F&S B&I WE ENA | ●● | [9, 25, 36, 104] |
| *Silene noctiflora* | N | ENA | | [5] |
| *Silene uniflora* | N | MED | | [60] |
| *Silene vulgaris* | N | F&S B&I | ● | [36, 63] |
| *Stellaria alsine* | N | M&NE | | [35] |
| *Stellaria graminea* | N,A | F&S B&I WE ENA | ● | [36, 63, 87, 88, 130] |
| *Stellaria media* | N,A | F&S B&I MED ENA | | [5, 6, 36, 37, 130] |
| *Stellaria nemorum* | N | WE | | [25] |
| *Stellaria palustris* | N | F&S | | [124] |
| **Celastraceae** | | | | |
| *Euonymus europaeus* | N | B&I | ●●● | |
| *Euonymus fortunei* | N | B&I | ●●● | |
| *Euonymus japonicus* | N | B&I MED | ●● | [75] |
| **Cercidiphyllaceae** | | | | |
| *Cercidiphyllum japonicum* | N | B&I | ●●● | |
| **Cistaceae** | | | | |
| *Cistus corbariensis* | N | B&I | ●●● | |
| *Cistus creticus* | N | MED | | [69] |
| *Cistus monspeliensis* | N,A | MED | | [106, 131] |
| *Cistus salvifolius* | N | MED | | [60] |
| *Helianthemum nummularium* | N | B&I WNA | ●● | [29] |
| **Commelinaceae** | | | | |
| *Commelina diffusa* | N | HI | | [28] |
| *Tradescantia fluminensi* | N | WNA | | [29] |
| **Convolvulaceae** | | | | |
| *Calystegia sepium* | N | B&I WE | ●● | [35] |
| *Calystegia silvatica* | N | NZ | ●● | [37] |
| *Convolvulus acicularis* | N | B&I MED | | [63, 97] |
| *Convolvulus arvensis* | N | B&I MED ENA WNA | ● | [5, 29, 40, 63] |
| *Convolvulus cneorum* | N | B&I | ●●● | |
| *Ipomea batatas* | N | HI | | [28] |
| *Ipomoea indica* | N | HI | | [27] |
| *Ipomoea muricata* | N | WNA | | [29] |
| **Coriariaceae** | | | | |
| *Coriaria* sp. | A | NZ | | [132] |

*(Continued)*

**Table 1.** (Continued)

| Host species by family | Stage | Geographic area(s) | Cit. sci. obs. | Selected record source(s) |
|---|---|---|---|---|
| **Cornaceae** | | | | |
| *Cornus alba* | A | EorWNA | | [126] |
| *Cornus canadensis* | N | ENA | | [87] |
| *Cornus racemosa* | N | ENA | | [58] |
| *Cornus sanguinea* | A | MED | | [40] |
| *Cornus suecia* | N | F&S | | [36] |
| **Crassulaceae** | | | | |
| *Crassula sarcocaulis* | N | B&I | ●●● | |
| *Hylotelephium spectabile* | N | B&I | | J. Owen |
| *Petrosedum rupestre* | N | B&I | | J. Owen |
| *Phedimus spurius* | N | B&I | | J. Owen |
| *Sedum acre* | N | B&I | ●●● | |
| *Sedum album* | N | B&I | ●●● | |
| *Umbilicus oppositifolius* | N | B&I | ●●● | |
| *Umbilicus repestris* | N | MED | | [60] |
| **Cucurbitaceae** | | | | |
| *Cucumis sativus* | N | B&I | ●●● | |
| *Cucurbita pepo* | N | B&I | ●●● | |
| **Cupressaceae** | | | | |
| *Cupressus* sp. | A | MED | | [74] |
| *Hesperocyparis macrocarpa* | A | WNA | | VT-CA-2018-A |
| *Juniperus communis* | A | B&I | | [133] |
| *Juniperus oxycedrus* | A | MED | | [7] |
| *Juniperus virginiana* | A | ENA | | VT-CT-2004-A |
| **Cyperaceae** | | | | |
| *Carex echinata* | N | B&I | ●●● | |
| *Carex hirta* | N | WE | | [25] |
| *Carex nigra* | N | F&S | | [134] |
| *Carex panacea* | N | B&I | ●●● | |
| *Kyllinga brevifolia* | N | HI | | [28] |
| *Schoenoplectus tabernaemontani* | N | F&S | | [134] |
| *Scirpus* sp. | N | ENA | | [58] |
| **Elaeagnaceae** | | | | |
| *Hippophae rhamnoides* | N | B&I WE | | [135] JRpercom |
| **Elaeocarpaceae** | | | | |
| *Crinodendron hookerianum* | N | B&I | ●●● | |
| **Equisetaceae** | | | | |
| *Equisetum arvense* | N | F&S B&I MED ENA | ●● | [58, 86, 121] |
| *Equisetum silvaticum* | N | F&S | | [36] |
| **Ericaceae** | | | | |
| *Andromeda polifolia* | N | B&I | ●●● | |
| *Arbusto* sp. | A | MED | | [40] |
| *Arbutus unedo* | N | B&I MED | ●● | [69] |
| *Calluna vulgaris* | N,A | F&S B&I | ● | [36, 121, 136, 137] |
| *Enkianthus* sp. | A | EorWNA | | [126] |
| *Erica cinerea* | N | B&I | ●●● | |
| *Erica scoparia* | A | AZ | | [138] |

(*Continued*)

**Table 1.** (Continued)

| Host species by family | Stage | Geographic area(s) | Cit. sci. obs. | Selected record source(s) |
|---|---|---|---|---|
| *Erica tetralix* | N | B&I | ●●● | |
| *Erica x darleyensis* | N | B&I | ●●● | |
| *Leptecophylla tameiameiae* | N | HI | | [27] |
| *Oxydendrum arboreum* | N | B&I | ●●● | |
| *Pyrola minor* | N | F&S | | [36] |
| *Vaccinium angustifolium* | N,A | ENA | | [87, 139] |
| *Vaccinium cylindraceum* | N | AZ | | [53] |
| *Vaccinium macrocarpon* | N,A | WNA | | [140] |
| *Vaccinium myrtillus* | N | F&S B&I | ●● | [36] |
| *Vaccinium ovatum* | N | B&I | ●●● | |
| *Vaccinium reticulatum* | N | HI | | [27] |
| *Vaccinium uliginosum* | N | F&S | | [36] |
| *Vaccinium vitis-idaea* | N | F&S | | [36] |
| **Escalloniaceae** | | | | |
| *Escallonia* sp. | N | B&I | | HMBL-England |
| **Euphorbiaceae** | | | | |
| *Euphorbia characias* | N | B&I | ●●● | |
| *Euphorbia cyparissias* | N | WE MED | | [25, 131] |
| *Euphorbia dulcis* | N | WE | | [35] |
| *Euphorbia griffithii* | N | B&I | ●●● | |
| *Euphorbia purpurea* | N | B&I | ●●● | |
| *Euphorbia serrata* | N | WE | | [83] |
| *Euphorbia terracina* | N | MED | | [60] |
| *Mercurialis perennis* | N | B&I | ● | [37] |
| *Ricinus communis* | N | B&I | ●●● | |
| **Fabaceae** | | | | |
| *Acacia longifolia* | N | MED | | [60] |
| *Anthyllis vulneraria* | N | B&I | ●●● | |
| *Arachis hypogaea* | A | MED | | [85] |
| *Argyrolobium biebersteinii* | N | MED | | [75] |
| *Astragalus* sp. | N | MED | | [8] |
| *Baptisia australis* | N | B&I | ●●● | |
| *Bituminaria* sp. | N | MED | | [10] |
| *Colutea sp.* | A | EorWNA | | [126] |
| *Coronilla emerus* | A | WE | | [67] |
| *Coronilla scorpioides* | N | MED | | [11] |
| *Cytisus laniger* | N | MED | | [69] |
| *Cytisus multiflora* | N | MED | | [60] |
| *Cytisus praecox* | N | B&I | ●●● | |
| *Cytisus scoparius* | N,A | B&I MED WNA | ●● | [60] VT-OR-2003-A |
| *Cytisus × kewensis* | N | B&I | | J. Owen |
| *Echinocystis fabacea* | N | WNA | | [29] |
| *Galega officinalis* | N | MED | | [75] |
| *Genista* sp. | A | EorWNA | | [126] |
| *Glycine max* | N | ENA | | [141] |
| *Hippocrepis* sp. | N | MED | | [10] |
| *Lathyrus japonicus* | N | ENA | | VT-ME-2021-N |

(*Continued*)

**Table 1.** (Continued)

| Host species by family | Stage | Geographic area(s) | Cit. sci. obs. | Selected record source(s) |
|---|---|---|---|---|
| *Lathyrus latifolius* | A | WNA | | VT-OR-2001-A |
| *Lathyrus linifolius* | N | B&I | ●●● | |
| *Lathyrus nissolia* | N | B&I | ●●● | |
| *Lathyrus ochrus* | N | MED | | [11] |
| *Lathyrus odoratus* | N | ENA | ●● | [5] |
| *Lathyrus palustris* | N | F&S | | [134] |
| *Lathyrus pratensis* | N | F&S B&I | ● | [36, 37] |
| *Lathyrus sativa* | N | B&I | ●●● | |
| *Lathyrus silvestris* | N | F&S | | [36] |
| *Lathyrus vernus* | N | F&S | | [36] |
| *Lotus angustissimus* | N | MED | | [11] |
| *Lotus corniculatus* | N | F&S B&I WE MED ENA NZ | ● | [9, 36, 37, 60, 88, 142] |
| *Lotus tetragonolobus* | N | B&I | | J. Owen |
| *Lupinus arboreus* | N | B&I | ●●● | |
| *Medicago cilaris* | N | MED | | [11] |
| *Medicago littoralis* | N | MED | | [60] |
| *Medicago lupulina* | N | B&I WE | | [26, 63] |
| *Medicago polymorpha* | N | MED ENA HI | | [5, 6, 28] |
| *Medicago rigidula* | N | MED | | [11] |
| *Medicago sativa* | N,A | MED ENA WNA NZ | | [5, 29, 70, 143, 144] |
| *Medicago scutellata* | N | MED | | [11] |
| *Melilotus alba Desv.* | N | ENA | | [145] |
| *Melilotus indica* | N | WNA | | [29] |
| *Melilotus officinalis* | N | F&S MED ENA | | [5, 36, 75] |
| *Onobrychis viciifolia* | N,A | WE EE MED | | [74, 83, 146] |
| *Ononis arvensis* | N | F&S | | [36] |
| *Ononis repens* | N | B&I | ●●● | |
| *Ononis spinosa* | N | B&I | ●●● | |
| *Ornithopus compressus* | N | MED | | [92] |
| *Phaseolus coccineus* | N | B&I | ●●● | |
| *Phaseolus vulgaris* | A | ENA | | [141] |
| *Pisum sativum* | N,A | B&I ENA | ●● | [32] |
| *Psorolea bituminosa* | N | MED | | [75] |
| *Robinia pseudoacacia* | A | MED | | [40] |
| *Scorpiurus* sp. | N | MED | | [10] |
| *Securigara* sp. | N | MED | | [10] |
| *Senna corymbosa* | N | B&I | ●●● | |
| *Trifolium alexandrinum* | N | MED | | [11] |
| *Trifolium arvense* | N | WE | | [25] |
| *Trifolium campestre* | N | HI | | [28] |
| *Trifolium hybridum* | N | F&S WE ENA | | [5, 26, 36] |
| *Trifolium medium* | N | F&S ENA | | [5, 36] |
| *Trifolium pratense* | N,A | F&S B&I WE MED ENA NZ | ● | [9, 36, 37, 43, 147] |
| *Trifolium repens* | N | F&S B&I MED ENA NZ | ● | [5, 6, 36, 63, 148] |
| *Trifolium rubens* | N | B&I | ●●● | |
| *Trifolium spadiceum* | N | F&S | | [36] |
| *Ulex europeaus* | N,A | B&I | ● | [149] |

(*Continued*)

**Table 1.** (Continued)

| Host species by family | Stage | Geographic area(s) | Cit. sci. obs. | Selected record source(s) |
|---|---|---|---|---|
| *Vicia cracca* | N | F&S B&I | ●● | [36] |
| *Vicia faba* | N | MED | | [9] |
| *Vicia hirsuta* | N | B&I WE | ● | [37, 82] |
| *Vicia melanops* | N | MED | | [11] |
| *Vicia sativa* | N | B&I WE MED ENA WNA NZ | ● | [25, 32, 37, 60, 110, 131] |
| *Vicia sepium* | N | F&S B&I WE | ● | [35–37] |
| *Vicia tetrasperma* | N | B&I | ●●● | |
| *Vicia villosa Roth* | N,A | MED ENA WNA | | [11, 33] SEMC |
| **Fagaceae** | | | | |
| *Castanea sativa* | N,A | B&I MED | ●● | [44, 74] |
| *Fagus sylvatica* | N | B&I | ●●● | |
| *Quercus agrifolia* | A | WNA | | VT-CA-2018-A |
| *Quercus cerris* | A | MED | | [150] |
| *Quercus crenata* | A | MED | | [10] |
| *Quercus ilex* | A | MED | | [7, 10, 22] |
| *Quercus infectoria* | A | MED | | [150] |
| *Quercus petraea.* | A | MED | | [40] |
| *Quercus pubescens* | A | MED | | [45] |
| *Quercus robur* | N,A | B&I WE | ●● | [67] |
| *Quercus trojana* | A | MED | | [10] |
| **Gentianaceae** | | | | |
| *Centaurium erythraea* | N | B&I WE HI | ● | [28, 63, 82] |
| *Gentiana asclepiadea* | N | M&NE | | [35] |
| *Menyanthes trifoliata* | N | F&S B&I | ●● | [36] |
| **Geraniaceae** | | | | |
| *Erodium cicutarium* | N | MED | | [6, 11, 92] |
| *Geranium carolinianum* | N | HI | | [28] |
| *Geranium cinereum* | N | B&I | | J. Owen |
| *Geranium dissectum* | N,A | B&I MED | ●● | [60] |
| *Geranium macrorrhizum* | N | B&I | ●●● | |
| *Geranium maculatum* | N | B&I | ●●● | |
| *Geranium molle* | N | MED | ●● | [6, 92] |
| *Geranium platypetalum* | N | B&I | ●●● | |
| *Geranium pratense* | N | B&I | ●●● | |
| *Geranium pusillum* | N | B&I WE | ●● | [35] |
| *Geranium pyrenaicum* | N | B&I | ●●● | |
| *Geranium robertianum* | N | B&I WE MED | ●● | [35, 60] |
| *Geranium sanguineum* | N | F&S, B&I | | [124] HMBL-England |
| *Geranium sylvaticum* | N | F&S | | [36] |
| *Pelargonium graveolens* | N | B&I WNA HI | ●● | [27, 29] |
| *Pelargonium peltatum* | N | WNA | | [29] |
| *Pelargonium × domesticum* | N | WNA | | [29] |
| *Pelargonium × hortorum* | N | WNA | | [29] |
| **Grossulariaceae** | | | | |
| *Ribes nigrum* | N | F&S B&I | ●● | [36] |
| *Ribes rubrum* | N | B&I WNA | ●● | [29] |
| *Ribes sanguineum* | N | B&I WE | ●● | [35] |

(Continued)

**Table 1.** (Continued)

| Host species by family | Stage | Geographic area(s) | Cit. sci. obs. | Selected record source(s) |
|---|---|---|---|---|
| *Ribes uva-crispa* | N | B&I WE | ●● | [35] |
| **Hydrangeaceae** | | | | |
| *Hydrangea macrophylla* | N | WE | | [25] |
| *Hydrangea paniculata* | N | B&I WNA | ●● | [29] |
| *Hydrangea petiolaris* | N | B&I | ●●● | |
| *Philadelphus coronarius* | N | B&I | | J. Owen |
| **Hypericaceae** | | | | |
| *Hypericum androsaemum* | N | B&I | ●●● | |
| *Hypericum calycinum* | N | B&I | ●●● | |
| *Hypericum hirsutum* | N | B&I | | [63] |
| *Hypericum maculatum* | N | F&S | | [36] |
| *Hypericum moserianum* | N | WNA HI | | [28, 29] |
| *Hypericum olympicum* | N | B&I | ●●● | |
| *Hypericum perforatum* | N,A | F&S B&I WE MED | ● | [7, 26, 36, 37, 63] |
| *Hypericum pulchrum* | N | B&I | ●●● | |
| *Hypericum tetrapterum* | N | F&S, B&I | ●● | [124] |
| **Iridaceae** | | | | |
| *Crocosmia × crocosmiiflora* | N | B&I, HI | | [28] HMBL-England |
| *Gladiolus communis* | N | B&I | ●●● | |
| *Hesperantha coccinea* | N | B&I | ●●● | |
| *Iris germanica* | N | MED | | [75] |
| *Iris pseudacorus* | N | B&I | ●●● | |
| *Iris sibirica* | N | B&I | ●●● | |
| *Iris versicolor* | N | ENA | | [87] |
| *Libertia ixioides* | N | B&I | ●●● | |
| **Juglandaceae** | | | | |
| *Carya ovata* | N | ENA | | [5] |
| *Juglans nigra* | A | ENA | | [151] |
| *Juglans regia* | ? | MED | | [152] |
| **Juncaceae** | | | | |
| *Juncus acutiflorus* | N | B&I | ●●● | |
| *Juncus acutus* | N | B&I | ●●● | |
| *Juncus effusus* | N | B&I | ● | [37] |
| *Juncus inflexus* | N | B&I | ●●● | |
| *Juncus gerardii* | N | F&S | | [134] |
| *Juncus squarrosus* | N | B&I | ●●● | |
| **Juncaginaceae** | | | | |
| *Triglochin maritima* | N | F&S | | [36] |
| **Lamiaceae** | | | | |
| *Agastache foeniculum* | N | B&I | ●●● | |
| *Agastache rugosa* | N | B&I | ●●● | |
| *Ajuga reptans* | N | B&I ENA | ● | [5, 63] |
| *Callicarpa* sp. | A | EorWNA | | [126] |
| *Dracocephalum parviflorum* | N | ENA | | [5] |
| *Galeopsis bifida* | N | F&S | | [36] |
| *Galeopsis speciosa* | N | F&S | | [36] |
| *Galeopsis tetrahit* | N | B&I WE | | [35, 37] |

(*Continued*)

**Table 1.** (Continued)

| Host species by family | Stage | Geographic area(s) | Cit. sci. obs. | Selected record source(s) |
|---|---|---|---|---|
| *Glechoma hederacea* | N | F&S B&I ENA | ●● | [5, 36] |
| *Hyssopus officinalis* | N | B&I WE | ● | [115, 153] |
| *Lamium album* | N | WE | | [35] |
| *Lamium amplexicaule* | N | ENA | | [5] |
| *Lamium galeobdolon* | N | B&I | ●●● | |
| *Lamium maculatum* | N | WE MED | | [60, 82] |
| *Lavandula angustifolia* | N,A | B&I MED | ●● | [154] |
| *Lavandula stoechas* | N,A | B&I MED | ●● | [7, 69] |
| *Lavandula x intermedia* | N | B&I | | HMBL-England |
| *Lycopus europaeus* | N | F&S | | [36] |
| *Melissa officinalis* | N | B&I | ●●● | |
| *Mentha aquatica* | N | B&I M&NE | ●● | [35] |
| *Mentha arvensis* | N | B&I M&NE | ●● | [35] |
| *Mentha spicata* | N | B&I | ●●● | |
| *Mentha suaveolens* | N | B&I | ●●● | |
| *Mentha x gracilis* | N | B&I | ●●● | |
| *Mentha x piperita* | N | B&I WE | ●● | [155] |
| *Mentha spicata* | N | WNA | | [29] |
| *Monarda fistulosa* | N | ENA | | [5] |
| *Nepeta cataria* | N | B&I ENA | ●● | [5] |
| *Nepeta racemosa* | N | B&I WNA | | J. Owen [29] |
| *Nepeta x faassenii* | N | B&I | ●●● | |
| *Ocimum basilicum* | N | B&I | ●●● | |
| *Origanum majorana* | N | B&I MED | ●● | [29] |
| *Origanum vulgare* | N | B&I WNA | ●●● | VT-CA-2021-N |
| *Physostegia virginiana* | N | B&I | ●●● | |
| *Prunella vulgaris* | N | F&S B&I WE | | [35, 36, 63] |
| *Salvia coccinea* | N | WNA | | [29] |
| *Salvia elegans* | N | B&I | ●●● | |
| *Salvia farinacea* | N | B&I | ●●● | |
| *Salvia guaranitica* | N | B&I | ●●● | |
| *Salvia longispicata* | N | B&I | ●●● | |
| *Salvia mircophylla* | N | B&I | ●●● | |
| *Salvia nemorosa* | N | B&I | ●●● | |
| *Salvia officialis* | N | B&I WNA | ●● | [29] |
| *Salvia rosmarinus* | N | B&I MED WNA | ● | [8, 29, 120] |
| *Salvia splendens* | N | WE | | [25] |
| *Salvia x jamensis* | N | B&I | ●●● | |
| *Salvia yangii* | N | B&I | ●●● | |
| *Satureja hortensis* | N | B&I | ●●● | |
| *Satureja montana* | N | B&I | ●●● | |
| *Scutellaria galericulata* | N | F&S | | [36] |
| *Stachys ajugoides* | N | WNA | | [29] |
| *Stachys bullata* | N | WNA | | [29] |
| *Stachys byzantina* | N | B&I | ●●● | JRpercom |
| *Stachys germanica* | N | B&I | | [156] |
| *Stachys palustris* | N | M&NE | | [35] |

(Continued)

**Table 1.** (Continued)

| Host species by family | Stage | Geographic area(s) | Cit. sci. obs. | Selected record source(s) |
|---|---|---|---|---|
| *Stachys rigida* | N | WNA | | RKpercom |
| *Stachys sylvatica* | N | B&I WE | | [35, 37] |
| *Teucrium scorodonia* | N | B&I | ●●● | |
| *Thymus citriodorus* | N | B&I | ●●● | |
| *Thymus mastichina* | A | MED | | [7] |
| *Thymus serpyllum* | N | B&I | ●●● | |
| *Thymus vulgaris* | N | F&S WNA | ●● | [29] |
| **Lauraceae** | | | | |
| *Laurus nobilis* | N | B&I | ●●● | |
| **Liliaceae** | | | | |
| *Lilium martagon* | N | B&I | ●●● | |
| *Tulipa* sp. | N | B&I | | J. Owen |
| **Linaceae** | | | | |
| *Linum catharticum* | N | B&I | ●●● | |
| *Linum usitatissimum* | N,A | F&S B&I | ●● | [157] |
| **Lythraceae** | | | | |
| *Lythrum maritimum* | N | ENA HI | | [28, 32] |
| *Lythrum salicaria* | N | F&S B&I M&NE ENA | ●● | [35, 36, 158] |
| **Magnoliaceae** | | | | |
| *Magnolia denudata* | N | B&I | ●●● | |
| *Magnolia liliiflora* | N | B&I | ●●● | |
| *Magnolia stellata* | N | B&I | ●●● | |
| **Malvaceae** | | | | |
| *Alcea rosea* | N | B&I, WNA | | [29] J. Owen |
| *Althaea cannabina* | N | B&I | ●●● | |
| *Althaea officinalis* | ? | WE | | [72] |
| *Gossypium hirsutum* | ? | EE | | [146] |
| *Hibiscus syriacus* | N | B&I | ●●● | |
| *Hibiscus tiliaceus* | N | HI | | [28] |
| *Kokia drynarioides* | N | HI | | [27] |
| *Malva moschata* | N | B&I | ● | [63] |
| *Malva neglecta* | N | WE | | [25] |
| *Malva parviflora* | N | WNA | | [29] |
| *Malva sylvestris* | N | B&I MED | ●● | [11] |
| *Modiola caroliniana* | N | HI | | [28] |
| *Tilia platyphyllos* | N | F&S | | [121] |
| **Melastomataceae** | | | | |
| *Tibouchina semidecandra* | N | HI | | [28] |
| **Montiaceae** | | | | |
| *Claytonia virginica* | N | ENA | | [5] |
| **Moraceae** | | | | |
| *Ficus carica* | N,A | B&I WE | ●● | [36, 67, 72] |
| *Morus alba* | A | WE | | [67] |
| **Musaceae** | | | | |
| *Musa* sp. | A | MED | | [85] |
| **Myricaceae** | | | | |
| *Comptonia peregrina* | A | ENA | | VT-WI-2000-A |

(*Continued*)

**Table 1.** (Continued)

| Host species by family | Stage | Geographic area(s) | Cit. sci. obs. | Selected record source(s) |
|---|---|---|---|---|
| *Myrica faya* | A | AZ | | [138] |
| *Myrica gale* | N | F&S B&I | ●● | [36, 137] |
| **Myrtaceae** | | | | |
| *Calothamnus graniticus* | N | B&I | ●●● | |
| *Eucalyptus amygdalina* | N | B&I | ●●● | |
| *Leptospermum laevigatum* | N | WNA | | [29] |
| *Leptospermum scoparium* | N | B&I | ●●● | |
| *Metrosideros collina* | N | HI | | [28] |
| *Myrtus communis* | N,A | B&I MED | ●● | [68] |
| **Nartheciaceae** | | | | |
| *Narthecium ossifragum* | N | B&I | | HMBL-Scotland |
| **Nymphaeaceae** | | | | |
| *Nymphaea* sp. | N | B&I | | [39] |
| **Oleaceae** | | | | |
| *Forsythia suspensa* | N | B&I | ●●● | |
| *Forsythia × intermedia* | N | B&I | | J. Owen |
| *Fraxinus excelsior* | N | F&S B&I | ●● | [121] |
| *Fraxinus ornus* | A | MED | | [10] |
| *Fraxinus pennsylvanica* | N | ENA | | [5] |
| *Jasminum humile* | N | B&I | ●●● | |
| *Jasminum nudiflorum* | N | B&I | ●●● | |
| *Jasminum officinale* | N | B&I | ●●● | |
| *Ligustrum ovalifolium* | N | B&I ENA | ●● | [5] |
| *Ligustrum vulgare* | N | B&I ENA | ●● | [5] |
| *Olea europaea* | N,A | B&I MED | ●● | [6, 7, 78, 110, 159] |
| *Osmanthus delavayi* | N | B&I | ●●● | |
| *Syringa vulgaris* | N | F&S B&I ENA | ●● | [5, 36] |
| **Onagraceae** | | | | |
| *Chamaenerion angustifolium* | N,A | F&S B&I WE ENA WNA | ● | [36, 87, 91, 160] VT-WA-2003-N |
| *Circaea canadensis* | N | B&I | ●●● | |
| *Epilobium alsinifolium* | N | B&I | ●●● | |
| *Epilobium ciliatum* | N | B&I | ●●● | |
| *Epilobium hirsutum* | N | B&I | ●●● | |
| *Epilobium lanceolatum* | N | B&I | ●●● | |
| *Epilobium montanum* | N | F&S B&I WE | ● | [35–37] |
| *Epilobium obscurum* | N | B&I M&NE | ●● | [35] |
| *Epilobium parviflorum* | N | B&I M&NE | ●● | [35] |
| *Epilobium roseum* | N | M&NE | | [35] |
| *Epilobium tetragonum* | N | B&I | ●●● | |
| *Fuchsia arborescens* | N | HI | | [27] |
| *Fuchsia magellanica* | N | B&I HI | ●● | [28] |
| *Fuchsia microphylla* | N | B&I | ●●● | |
| *Fuchsia procumbens* | N | B&I | ●●● | |
| *Fuchsia triphylla* | N | WNA | | [29] |
| *Oenothera biennis* | N,A | B&I, WE ENA | ●● | [26, 161] HMBL-England |
| *Oenothera glaziovana* | N | NZ | | [37] |
| *Oenothera lindheimeri* | N | B&I | ●●● | |

*(Continued)*

**Table 1.** (Continued)

| Host species by family | Stage | Geographic area(s) | Cit. sci. obs. | Selected record source(s) |
|---|---|---|---|---|
| *Oenothera odorata* | N | HI | | [42] |
| *Oenothera stricta* | N | HI | | [28] |
| **Orchidaceae** | | | | |
| *Dactylorhiza fuchsii* | N | B&I | ●●● | |
| *Dactylorhiza maculata* | N | B&I | ●●● | |
| *Neottia ovata* | N | B&I | | [37] |
| **Orobanchaceae** | | | | |
| *Bellardia* sp. | N | MED | | [8] |
| *Euphrasia officinalis* | N | B&I | | CH-England-2019-N |
| *Melampyrum nemorosum* | N | F&S | | [36] |
| *Melampyrum pratense* | N | F&S, MED | | [36] HMBL-Italy |
| *Melampyrum sylvaticum* | N | F&S | | [36] |
| *Odontites vernus* | N | B&I | | CH-England-2019-N |
| *Pedicularis palustris* | N | WE | | [88] |
| *Rhinanthus minor* | N | F&S B&I WE ENA | ●● | [35, 36, 87] |
| *Rhinanthus serotinus* | N | F&S | | [36] |
| **Osmundaceae** | | | | |
| *Osmunda regalis* | N | B&I | ●●● | |
| **Oxalidaceae** | | | | |
| *Oxalis acetosella* | N | WE | | [88] |
| *Oxalis corniculata* | N | HI | | [28] |
| *Oxalis stricta* | N | ENA | | [104] |
| **Papaveraceae** | | | | |
| *Chelidonium majus* | N | WE | | [25] |
| *Eschscholzia californica* | N | B&I WNA | ●● | [29] |
| *Fumaria* sp. | N | MED | | [10] |
| *Papaver cambricum* | N | B&I | ●●● | |
| *Papaver dubium* | N | WE | | [26] |
| *Papaver nudicaule* | N | B&I | ●●● | |
| *Papaver orientale* | N | B&I ENA WNA | ●● | [5, 29] |
| *Papaver rhoeas* | N | B&I WNA | ●● | [29] |
| *Papaver somniferum* | N | B&I WE | ●● | [25] |
| *Roemeria argemone* | N | WE | | [26] |
| *Stylophorum diphyllum* | N | B&I | ●●● | |
| **Phrymaceae** | | | | |
| *Diplacus aurantiacus* | N | B&I | ●●● | |
| *Erythranthe guttata* | N | WNA | | [162] |
| *Erythranthe lutea* | N | B&I | ●●● | |
| **Phyllanthaceae** | | | | |
| *Phyllanthus niruri* | N | B&I | ●●● | |
| **Pinaceae** | | | | |
| *Abies balsamea* | N | ENA | | [87] |
| *Picea glauca* | N | ENA | | [87] |
| *Pinus banksiana* | A | ENA | | VT-MI-2001-A |
| *Pinus contorta* | N,A | B&I WNA | | [137] VT-CA-2018-A |
| *Pinus halepensis* | A | MED | | [45] |
| *Pinus ponderosa* | N,A | WNA | | [163] VT-OR-2003-A |

(*Continued*)

**Table 1.** (Continued)

| Host species by family | Stage | Geographic area(s) | Cit. sci. obs. | Selected record source(s) |
|---|---|---|---|---|
| *Pinus radiata* | A | WNA | | VT-CA-2018-A |
| *Pinus resinosa* | A | ENA | | VT-MI-2001-A |
| *Pinus strobus* | N,A | ENA | | [33] VT-NH-2020-N |
| *Pinus sylvestris* | A | ENA | | [164] |
| *Pinus virginiana* | A | ENA | | VT-MA-2006-A |
| **Pittosporaceae** | | | | |
| *Billardiera heterophylla* | N | B&I | ●●● | |
| *Pittosporum tenuifolium* | N | B&I | ●●● | |
| *Pittosporum undulatum* | A | AZ | | [138] |
| **Plantaginaceae** | | | | |
| *Antirrhinum majus* | N | B&I | | J. Owen |
| *Hebe rakaiensis* | N | B&I | ●●● | |
| *Hebe salicifolia* | N | B&I HI | ●● | [28] |
| *Linaria vulgaris* | N | F&S, WE | | [35, 124] |
| *Plantago coronopus* | N | MED | | [69] |
| *Plantago lagopus* | N | WE MED | | [11, 25] |
| *Plantago lanceolata* | N,A | B&I WE MED ENA WNA HI NZ | ● | [25, 28, 29, 32, 37, 40, 60, 63] |
| *Plantago major* | N,A | F&S B&I WE ENA NZ | ●● | [5, 25, 36, 37] |
| *Plantago media* | N | B&I WE | ●● | [25] |
| *Plantago maritima* | N | F&S B&I | ●● | [36] |
| *Plantago rugelii* | ? | ENA | | [33] |
| *Veronica agrestis* | N | WE | | [35] |
| *Veronica arvensis* | N | B&I WE MED | ●● | [25, 40] |
| *Veronica beccabunga* | N | M&NE | | [35] |
| *Veronica chamaedrys* | N | F&S B&I WE | ● | [35, 36, 63] |
| *Veronica hederifolia* | N | B&I | | J. Owen |
| *Veronica longifolia* | N | M&NE | | [35] |
| *Veronica officinalis* | N | WE | | [35] |
| *Veronica peduncularis* | N | B&I | ●●● | |
| *Veronica pinguifolia* | N | B&I | | HMBL-England |
| *Veronica peregrina* | N | ENA | | [5] |
| *Veronica persica* | N | B&I | | [63] |
| *Veronica plebeia* | N | HI | | [28] |
| *Veronica serpyllifolia* | N | B&I | ●●● | |
| *Veronica spicata* | N | B&I | | J. Owen |
| *Veronica urticifolia* | N | MED | | HMBL-Italy |
| *Veronicastrum* sp. | N | B&I | | HMBL-England |
| **Platanaceae** | | | | |
| *Platanus occidentalis* | N | ENA | | [5] |
| **Plumbaginaceae** | | | | |
| *Armeria maritima* | N | B&I | ●●● | |
| **Poaceae** | | | | |
| *Agrostis canina* | N | B&I | | [37] |
| *Agrostis capillaris* | N | B&I, WE MED | ● | [26, 37, 75] |
| *Agrostis gigantea* | N,A | B&I ENA | | [32, 37] |
| *Agrostis stolonifera* | N | B&I ENA NZ | ● | [5, 37] |
| *Alopecurus myosuroides* | N | B&I | ●●● | |

(*Continued*)

**Table 1.** (Continued)

| Host species by family | Stage | Geographic area(s) | Cit. sci. obs. | Selected record source(s) |
|---|---|---|---|---|
| *Alopercurus pratensis* | N | F&S B&I NZ | ●● | [36, 37] |
| *Anemanthele lessoniana* | N | B&I | ●●● | |
| *Anisantha* sp. | N | MED | | [10] |
| *Anthoxanthum odoratum* | N | B&I ENA NZ | ●● | [33, 37] |
| *Apera spica-venti* | N | WE | | [26] |
| *Arrhenatherum elatius* | N,A | B&I MED ENA NZ | ●● | [5, 37, 165] |
| *Arundo donax* | N | MED | | [9] |
| *Avena barbata* | N | MED | | [75] |
| *Avena fatua* | N | WNA | | [110] |
| *Avena sativa* | N,A | F&S ENA | | [5, 36, 157] |
| *Avena sterilis* | N | MED | | [6, 11] |
| *Brachiaria mutica* | N | HI | | [28] |
| *Brachypodium pinnatum* | N | WE | | [166] |
| *Bromus hordeaceus* | N | B&I WE MED NZ | | [11, 25, 37, 75] |
| *Bromus secalinus* | N | ENA | | [104] |
| *Bromus sterilis* | N | MED | | [6, 11] |
| *Bromus tectorum* | N | WE | | [26] |
| *Bromus willenowii* | N | NZ | | [37] |
| *Calamagrostis sp.* | N | F&S | | [36] |
| *Coix lachryma-jobi* | N | HI | | [28] |
| *Corynephorus canescens* | N | WE | | [26] |
| *Cyinosurus* sp. | ? | MED | | [97] |
| *Cynodon dactylon* | N | MED HI | | [28, 92] |
| *Cynosurus cristatus* | N | B&I NZ | ●● | [37] |
| *Dactylis glomerata* | N | B&I MED ENA WNA HI NZ | ● | [5, 6, 28, 37, 60] |
| *Dasypyrum villosum* | N | MED | | [11] |
| *Deschampsia cespitosa* | N | B&I | ●●● | |
| *Dichanthelium dichotomum* | N | ENA | | VT-NH-2020-N |
| *Digitaria horizontalis* | N | HI | | [28] |
| *Elymus repens* | N | B&I WE ENA NZ | ●● | [5, 25, 37] |
| *Festuca arundinacea* | N | F&S B&I NZ | ●● | [37, 134] |
| *Festuca glauca* | N | B&I WE | ●● | [26] |
| *Festuca ovina* | N | B&I | ●●● | [26] |
| *Festuca rubra* | N | B&I WE | ●● | [26] |
| *Helictotrichon pratense* | N | B&I | ●●● | |
| *Holcus lanatus* | N,A | B&I WE HI NZ | ● | [25, 28, 37, 167] |
| *Holcus mollis* | N | B&I WE NZ | ●● | [37, 72] |
| *Hordeum brachyantherum* | N | WNA | | [110] |
| *Hordeum bulbosum* | N | MED | | [75] |
| *Hordeum murinum* | N | MED WNA NZ | | [6, 29, 37, 75] |
| *Hordeum vulgare* | N,A | F&S ENA | | [5, 32, 36, 157] |
| *Lagurus ovatus* | N | MED | | [60] |
| *Leymus arenarius* | N | B&I | ●●● | |
| *Lolium multiflorum* | N,A | ENA | | [32] |
| *Lolium perenne* | N | B&I WE MED NZ | ● | [11, 26, 37] |
| *Melica uniflora* | N | B&I | ●●● | |
| *Milium effusum* | N | WE | | [88] |

(*Continued*)

**Table 1.** (Continued)

| Host species by family | Stage | Geographic area(s) | Cit. sci. obs. | Selected record source(s) |
|---|---|---|---|---|
| *Molinia caerulea* | N | B&I | ●●● | |
| *Muhlenbergia rigens* | N | B&I | ●●● | |
| *Nassella tenuissima* | N | B&I | ●●● | |
| *Panicum virgatum* | N | B&I | ●●● | |
| *Pennisetum clandestinum* | N | B&I | ●●● | |
| *Phalaris aquatica* | N | NZ | | [37] |
| *Phalaris arundinacea* | N | B&I | ●●● | |
| *Phleum pratense* | N | F&S B&I ENA NZ | ●● | [5, 36, 37] |
| *Phragmites australis* | N | F&S B&I WE MED | ●● | [88, 134, 168] |
| *Poa annua* | N | B&I MED | ●● | [6] |
| *Poa pratensis* | N | B&I WE ENA NZ | ● | [5, 26, 37] |
| *Poa trivialis* | N | B&I NZ | ● | [37] |
| *Rostraria* sp. | ? | MED | | [97] |
| *Sacciolepis indica* | N | HI | | [28] |
| *Secale cereale* | N | F&S WE | | [25, 36] |
| *Sorghum halepense* | N | Italy | | [98] |
| *x Triticosecale* | A | EE | | [169] |
| *Triticum aestivum* | N,A | F&S MED EorWNA | | [32, 36, 65, 98] |
| *Triticum monococcum* | N | EE | | [170] |
| *Triticum polonicum* | N | EE | | [170] |
| *Zea mays* | N,A | ENA | | [5, 32] |
| **Polemoniaceae** | | | | |
| *Phlox douglasii* | N | B&I | ●●● | |
| *Phlox paniculata* | N | B&I | ●●● | |
| *Phlox subulata* | N | B&I | ●●● | |
| *Polemonium caeruleum* | N | F&S, B&I | ●● | [124] |
| **Polygonaceae** | | | | |
| *Bistorta officinalis* | N | B&I WE | ●● | [35] |
| *Bistorta vivipara* | N | F&S | | [36] |
| *Fagopyrum esculentum* | A | ENA | | [58] |
| *Fallopia baldschuanica* | N | B&I | ●●● | |
| *Fallopia convovulus* | N | WE | | [35] |
| *Persicaria amphibia* | N | F&S, M&NE | | [35, 124] |
| *Persicaria lapathifolia* | N | B&I | ●●● | |
| *Persecaria maculosa* | N | B&I | ● | [37] |
| *Persicaria odorata* | N | B&I | ●●● | |
| *Polygonum aviculare* | N | F&S WE MED | | [9, 35, 36] |
| *Polygonum lapathifolium* | N | F&S | | [36] |
| *Reynoutria multiflora* | N | B&I | ●●● | |
| *Rheum rhaponticum* | N | B&I | | J. Owen |
| *Rheum x hybridum (?)* | N | B&I | ●● | [28, 171] |
| *Rumex acetosa* | N | F&S B&I WE MED | ● | [25, 26, 36, 60, 63] |
| *Rumex acetosella* | N | F&S B&I WE ENA WNA HI | ●● | [26, 28, 36, 87, 172] |
| *Rumex aquaticus* | N | F&S | | [36] |
| *Rumex bucephalophorus* | N | MED | | [6, 92] |
| *Rumex conglomeratus* | N | B&I WE WNA | ●● | [29, 35] |
| *Rumex crispis* | N | F&S, B&I WE MED ENA WNA | ●● | [9, 29, 87, 102, 124] |

(*Continued*)

**Table 1.** (Continued)

| Host species by family | Stage | Geographic area(s) | Cit. sci. obs. | Selected record source(s) |
|---|---|---|---|---|
| *Rumex hydrolapathum* | N | WE | | [35] |
| *Rumex induratus* | N | MED | | [92] |
| *Rumex longifolius* | N | F&S | | [36] |
| *Rumex occidentalis* | ? | ENA | | [33] |
| *Rumex obtusifolius* | N,A | B&I WE ENA | ● | [5, 25, 173] |
| *Rumex sanguineus* | N | B&I | ●●● | |
| **Polypodiaceae** | | | | |
| *Dryopteris carthusiana* | N | ENA | | [87] |
| *Ecballium elaterium* | N | MED | | [8] |
| *Pteridium aquilinum* | N,A | F&S B&I EE NZ | ● | [36, 174–176] |
| **Primulaceae** | | | | |
| *Anagallis arvensis* | N | MED HI | | [11, 28] |
| *Glaux maritima* | N | F&S | | [36] |
| *Lysimachia arvensis* | N | Italy | | [11] |
| *Lysimachia ciliata* | N | B&I | | HMBL-England |
| *Lysimachia clethroides* | N | B&I | ●●● | |
| *Lysimachia europaea* | N | F&S | | [36] |
| *Lysimachia foemina* | N | MED | | [6] |
| *Lysimachia nemorum* | N | B&I | | [37] |
| *Lysimachia nummularia* | N | B&I, WE | | [35] J. Owen |
| *Lysimachia punctata* | N | B&I | ●●● | |
| *Lysimachia thyrsiflora* | N | F&S | | [36] |
| *Lysimachia vulgaris* | N | F&S B&I WE | ●● | [36, 72] |
| *Primula auricula* | N | B&I | ●●● | |
| *Primula denticulata* | N | B&I | ●●● | |
| *Primula prolifera* | N | B&I | ●●● | |
| *Primula rosea* | N | WE | | [35] |
| *Primula veris* | N | B&I | ● | [15, 63] |
| *Primula vialii* | N | B&I | ●●● | |
| *Primula vulgaris* | N | B&I | ● | [37] |
| **Proteaceae** | | | | |
| *Adenanthos cuneatus* | N | B&I | ●●● | |
| *Grevillea rosmarinifolia* | N | B&I | ●●● | |
| *Lambertia rariflora* | N | B&I | ●●● | |
| **Ranunculaceae** | | | | |
| *Adonis* sp. | N | MED | | [8] |
| *Anemonoides sylvestris* | N | M&N Europe | | [35] |
| *Aquilegia canadensis* | N | ENA | | [5] |
| *Aquilegia chrysantha* | N | WNA | | [29] |
| *Aquilegia vulgaris* | N | B&I | ●●● | |
| *Caltha palustris* | N | F&S B&I WE | ● | [35, 36, 63] |
| *Clematis cirrhosa* | N | B&I | ●●● | |
| *Clematis montana* | N | B&I | ●●● | |
| *Clematis tangutica* | N | B&I | ●●● | |
| *Clematis vitalba* | N | B&I WE MED | ●● | [75, 83] |
| *Eriocapitella hupehensis* | N | B&I | ●●● | |
| *Nigella damascena* | N | B&I | ●●● | |

*(Continued)*

**Table 1.** (Continued)

| Host species by family | Stage | Geographic area(s) | Cit. sci. obs. | Selected record source(s) |
|---|---|---|---|---|
| *Ranunculus abortivus* | N | ENA | | [5] |
| *Ranunculus acris* | N | F&S B&I WE ENA | ●● | [25, 36, 83, 87] |
| *Ranunculus arvensis* | N | WE | | [88] |
| *Ranunculus auricomus* | N | B&I | ●●● | |
| *Ranunculus bulbosus* | N | B&I WE | ●● | [26] |
| *Ranunculus cymbalaria* | N | B&I | ●●● | |
| *Ranunculus flammula* | N | F&S B&I | ●● | [36] |
| *Ranunculus lingua* | N | B&I M&NE | ●● | [35] |
| *Ranunculus repens* | N,A | F&S B&I WE MED WNA | ● | [29, 35, 36, 60, 63] |
| *Ranunculus sceleratus* | N | B&I | ●●● | |
| *Thalictrum flavum* | N | M&NE | | [35] |
| *Trollius chinensis* | N | B&I | ●●● | |
| **Resedaceae** | | | | |
| *Reseda lutea* | N | WE | | [25] |
| **Rhamnaceae** | | | | |
| *Ceanothus thyrsiflorus* | N | B&I | | HMBL-England |
| *Rhamnus alaternus* | N,A | B&I MED | ●● | [10] |
| *Rhamnus cathartica* | A | ENA | | [177] |
| **Rosaceae** | | | | |
| *Agrimonia eupatoria* | N | B&I ENA | ● | [58, 63] |
| *Alchemilla glabra* | N | B&I | | [63] |
| *Alchemilla mollis* | N | B&I | ●●● | |
| *Alchemilla vulgaris* | N | F&S | | [36] |
| *Amelanchier spicata* | N | F&S | | [36] |
| *Argentina anserina* | N | B&I | ●●● | |
| *Aronia* sp. | ? | ENA | | [178] |
| *Aruncus sylvester* | N | B&I | | J. Owen |
| *Chaenomeles japonica* | N | WNA | | [29] |
| *Chaenomeles speciosa* | N | B&I | ●●● | |
| *Comarum palustre* | N | B&I | ●●● | |
| *Cotoneaster conspicuus* | N | B&I | ●●● | |
| *Cotoneaster franchetii* | N | B&I | ●●● | |
| *Cotoneaster horizontalis* | N | B&I | ●●● | |
| *Cotoneaster salicifolius* | N | B&I | ●●● | |
| *Crataegus monogyna* | N,A | B&I M&NE | ● | [35, 179] |
| *Cydonia oblonga* | N | B&I | ●●● | |
| *Dasiphora fruticosa* | N | B&I | ●●● | |
| *Filipendula ulmaria* | N,A | F&S B&I WE | ● | [35–37, 72] |
| *Filipendula vulgaris* | N | B&I | | HMBL-England |
| *Fragaria chiloensis* | N | WNA HI | | [27, 29] |
| *Fragaria moschata* | N | WE | | [35] |
| *Fragaria vesca* | N | F&S, B&I | ●● | [36] |
| *Fragaria virginiana* | N | ENA | | [180] |
| *Fragaria x ananassa* | N,A | F&S B&I WE ENA HI Reunion | | [5, 36, 42, 57] |
| *Geum aleppicum* | N | ENA | | [181] |
| *Geum coccineum* | N | B&I | ●●● | |
| *Geum quellyon* | N | B&I | ●●● | |

(*Continued*)

**Table 1.** (*Continued*)

| Host species by family | Stage | Geographic area(s) | Cit. sci. obs. | Selected record source(s) |
|---|---|---|---|---|
| *Geum rivale* | N | F&S B&I | ● | [36, 63] |
| *Geum ubanum* | N | F&S B&I WE MED | ● | [9, 25, 36, 37] |
| *Kerria japonica* | A | EorWNA | | [126] |
| *Malus domestica* | N,A | B&I WE ENA | ●● | [32, 58, 182] |
| *Malus sylvestris* | N,A | B&I ENA | ● | [183, 184] |
| *Margyricarpus pinnatus* | N | B&I | ●●● | |
| *Photinia x fraseri* | N | B&I | ●●● | |
| *Physocarpus opulifolius* | N | B&I | ●●● | |
| *Potentilla anserina* | N | F&S WE | | [35, 36] |
| *Potentilla argentea* | N | F&S | | [36] |
| *Potentilla canadensis* | N | ENA | | [5] |
| *Potentilla erecta* | N | F&S B&I WE | ●● | [25, 36, 137] |
| *Potentilla norvegica* | N | F&S, ENA | | [104, 124] |
| *Potentilla palustris* | N | F&S | | [36] |
| *Potentilla recta* | A | ENA | | [185] |
| *Potentilla reptans* | N | B&I | | [63] |
| *Potentilla sterilis* | N | B&I | ● | [37] |
| *Prunus americana* | ? | ENA | | [31] |
| *Prunus armeniaca* | N,A | ENA WNA | | [29, 186] |
| *Prunus avium* | N | B&I ENA | ●● | [58] |
| *Prunus cerasifera* | N | B&I | | J. Owen |
| *Prunus cerasus* | N,A | M&NE ENA | | [35, 187] |
| *Prunus domestica* | N,A | B&I WE MED ENA | ●● | [25, 74, 186] |
| *Prunus dulcis* | N,A | B&I MED | ●● | [7, 78] |
| *Prunus laurocerasus* | A | WE | | [188] |
| *Prunus padus* | N | F&S | | [36] |
| *Prunus pensylvanica* | ? | ENA | | [189] |
| *Prunus persica* | A | EE ENA | | [58, 182] |
| *Prunus serrulata* | N | B&I | | J. Owen |
| *Prunus spinosa* | N | B&I | ●●● | |
| *Prunus virginiana* | A | ENA | | [41] VT-NH-2006-A |
| *Pyracantha rogersiana* | N | B&I | ●●● | |
| *Pyrus communis* | N,A | WE MED ENA | | [58, 74, 82] |
| *Pyrus spinosa* | A | MED | | [45] |
| *Rosa arvensis* | N | B&I | ●●● | |
| *Rosa canina* | N | F&S B&I WE | ●● | [25, 36] |
| *Rosa chinensis* | N | ENA | | [32] |
| *Rosa foetida* | N | B&I | ●●● | |
| *Rosa gallica* | N | B&I | ●●● | |
| *Rosa hybrids* (domesticated) | N | F&S WE ENA | | [5, 36, 82] |
| *Rosa multiflora* | N | ENA | ●● | [126] |
| *Rosa nitida* | N | ENA | | [87] |
| *Rosa pimpinellifolia* | N | B&I | ●●● | |
| *Rosa rubiginosa* | N | B&I | | HMBL-England |
| *Rosa rugosa* | N | B&I ENA | ●● | [87] |
| *Rosa setigera* | N | B&I | ●●● | |
| *Rosa × damascena* | A | EE | | [190] |

(*Continued*)

**Table 1.** (Continued)

| Host species by family | Stage | Geographic area(s) | Cit. sci. obs. | Selected record source(s) |
|---|---|---|---|---|
| *Rubus allegheniensis* | N | ENA | | [5] |
| *Rubus arcticus* | N | F&S | | [36] |
| *Rubus argutus* | N | HI | | [28] |
| *Rubus armeniacus* | N | B&I | | [191] |
| *Rubus caesius* | N | B&I M&NE | ●● | [35] |
| *Rubus canadensis* | N,A | ENA | | [5, 32] |
| *Rubus canescens* | ? | MED | | [97] |
| *Rubus chamaemorus* | N | F&S B&I | ●● | [36] |
| *Rubus fruticosus* | N | B&I NZ | ● | [37] |
| *Rubus hispidus* | N | ENA | | [87] |
| *Rubus idaeus* | N | F&S B&I ENA | ●● | [36, 87] |
| *Rubus laciniatus* | N | B&I | | J. Owen |
| *Rubus occidentalis* | N,A | ENA | | [32] |
| *Rubus parviflorus* | N | WNA | | [29] |
| *Rubus procerus* | N | WNA | | [29] |
| *Rubus rosaceus* | ? | ENA | | [33] |
| *Rubus saxatilis* | N | F&S | | [36] |
| *Rubus ulmifolius* | N | B&I | ●●● | |
| *Rubus ursinus* | N | WNA | | VT-CA-2021-N |
| *Rubus vitifolius* | N | WNA | | [29] |
| *Rubus × loganobaccus* | N | B&I | ● | [191] |
| *Sanguisorba minor* | N | B&I MED | ●● | [97] |
| *Sanguisorba officinalis* | N | B&I | ●●● | |
| *Sanguisorba verrucosa* | N | MED | | [92] |
| *Sorbaria sorbifolia* | N | B&I | ●●● | |
| *Sorbus americana* | N | ENA | | [87] |
| *Sorbus aucuparia* | N | F&S B&I | ●● | [36] |
| *Spiraea alba* | N | ENA | | [102] |
| *Spiraea arguta* | N | B&I | ●●● | |
| *Spiraea cantoniensis* | N | WE | | [43] |
| *Spiraea douglasii* | N | B&I | ●●● | |
| *Spiraea japonica* | N | B&I | ●●● | |
| *Spiraea nipponica* | N | B&I | ●●● | |
| *Spiraea splendens* | N | WNA | | VT-WA-2003-N |
| *Spiraea vanhouttei* | N | B&I | | J. Owen |
| **Rubiaceae** | | | | |
| *Coprosma ernodeoides* | N | HI | | [28] |
| *Coprosma rhynchocarpa* | N | HI | | [28] |
| *Crucianella maritima* | N | MED | | [60] |
| *Cruciata laevipes* | N | B&I | ●●● | |
| *Galium aparine* | N | B&I WE MED ENA WNA | ● | [9, 29, 35, 37, 192] |
| *Galium asprellum* | N | ENA | | [104] |
| *Galium boreale* | N | F&S | | [36] |
| *Galium mollugo* | N | F&S B&I WE MED ENA | ●● | [9, 36, 88] VT-NH-2020-N |
| *Galium palustre* | N | F&S B&I | ●● | [36] |
| *Galium saxatile* | N | B&I | ●●● | |
| *Galium sylvaticum* | N | WE ENA | | [5, 35] |

*(Continued)*

**Table 1.** (Continued)

| Host species by family | Stage | Geographic area(s) | Cit. sci. obs. | Selected record source(s) |
|---|---|---|---|---|
| *Galium uliginosum* | N | F&S | | [36] |
| *Galium verum* | N | F&S B&I WE ENA | ●● | [5, 35, 36] |
| *Palicourea elata* | N | B&I | ●●● | |
| *Rubia peregrina* | N | B&I | ●●● | |
| *Rubia tinctorum* | N | B&I | | JRpercom |
| *Sherardia arvensis* | N | MED | | [10, 78] |
| *Theligonum* sp. | N | MED | | [10] |
| **Rutaceae** | | | | |
| *Citrus* sp. | A | MED | | [74] |
| *Phellodendron* sp. | A | EorWNA | | [126] |
| *Ruta* sp. | N | MED | | [10] |
| **Salicaceae** | | | | |
| *Populus nigra* | N | MED | | [75] |
| *Populus tremula* | N | F&S | | [36] |
| *Salix alba* | N,A | WE MED EorWNA | | [26, 67, 75, 126] |
| *Salix babylonica* | N,A | B&I MED | ●● | [75, 123] |
| *Salix caprea* | N,A | F&S B&I WE | ●● | [67, 121] |
| *Salix cinerea* | N | F&S | | [121] |
| *Salix fragilis* | N | WE | | [26] |
| *Salix phylicifolia* | N | F&S | | [36] |
| *Salix purpurea* | A | WE | | [67] |
| *Salix repens* | N | F&S | | [121] |
| *Salix viminalis* | N | F&S EE | | [121, 193] |
| **Santalaceae** | | | | |
| *Thesium* sp. | N | MED | | [40] |
| **Sapindaceae** | | | | |
| *Acer campestre* | N,A | B&I MED | ●● | [123] |
| *Acer negundo* | N | ENA | | VT-WI-1995-A |
| *Acer palmatum* | N | B&I | ●●● | |
| *Acer platanoides* | N | B&I | ●●● | |
| *Acer pseudoplatanus* | N | B&I | ●●● | |
| *Acer pycnanthum* | N | B&I | ●●● | |
| *Acer rubrum* | N | ENA | | VT-NH-2020-N |
| *Acer spicatum* | N | ENA | | [87] |
| *Aesculus hippocastanum* | N | B&I | ●●● | |
| *Aesculus parviflora* | A | EorWNA | | [126] |
| *Hippuris vulgaris* | N | B&I | ●●● | |
| **Sapotaceae** | | | | |
| *Sideroxylon* sp. | N | HI | | [27] |
| **Saururaceae** | | | | |
| *Houttuynia cordata* | N | B&I | | HMBL-England |
| **Saxifragaceae** | | | | |
| *Bergenia crassifolia* | N | B&I | | J. Owen |
| *Chrysosplenium oppositifolium* | N | B&I | ●●● | |
| *Heuchera sanguinea* | N | WNA | | [29] |
| *Tellima grandiflora* | N | B&I | ●●● | |
| **Scrophulariaceae** | | | | |

(*Continued*)

**Table 1.** (Continued)

| Host species by family | Stage | Geographic area(s) | Cit. sci. obs. | Selected record source(s) |
|---|---|---|---|---|
| *Buddleja alternifolia* | N | B&I | ●●● | |
| *Buddleia davidii* | N | B&I | ● | [194] |
| *Buddleja globosa* | N | B&I | ●●● | |
| *Linaria purpurea* | N | B&I | ●●● | |
| *Phygelius capensis* | N | B&I | ●●● | |
| *Scrophularia auriculata* | N | B&I | ●●● | |
| *Scrophularia californica* | N | WNA | | [29] |
| *Verbascum phoeniceum* | N | B&I | | HMBL-England |
| *Verbascum thapsus* | ? | ENA | | [33] |
| *Scrophularia nodosa* | N | WE | | [35] |
| **Solanaceae** | | | | |
| *Capsicum annuum* | N | B&I WE | ●● | [83] |
| *Capsicum chinense* | N | B&I | ●●● | |
| *Cestrum elegans* | N | WNA | | [29] |
| *Lycoperscion esculentum* | N | EE EorWNA HI | | [27, 56, 65] |
| *Nicotiana tabacum* | ? | EorWNA | | [65] |
| *Physalis alkekengi* | N | ENA | | [104] |
| *Physalis peruviana* | N | HI | | [28] |
| *Solanum dulcamara* | N,A | WE ENA | | [87, 195] AMNH |
| *Solanum lycopersicum* | N | B&I | ●●● | |
| *Solanum nigrum* | N | WE | | [195] |
| *Solanum tuberosum* | N,A | B&I EE MED Uzbekistan ENA | ●● | [56, 62, 196, 197] |
| **Staphyleaceae** | | | | |
| *Staphylea* sp. | A | EorWNA | | [126] |
| **Tamaricaceae** | | | | |
| *Tamarix tetrandra* | N | B&I | ●●● | |
| **Taxaceae** | | | | |
| *Taxus baccata* | N | B&I | ●●● | |
| **Thymelaeaceae** | | | | |
| *Daphne mezereum* | N | B&I | | J. Owen |
| *Dirca palustris* | A | EorWNA | | [126] |
| *Wikstroemia phillyreifolia* | N | HI | | [28] |
| **Tropaeolaceae** | | | | |
| *Tropaeolum majus* | N | B&I | ●●● | |
| **Ulmaceae** | | | | |
| *Celtis occidentalis* | A | ENA | | VT-IL-2001-A |
| *Ulmus glabra* | N | F&S | | [121] |
| *Ulmus laevis* | A | WE | | [67] |
| *Ulmus minor* | A | WE MED | | [45, 67] |
| **Urticaceae** | | | | |
| *Parietaria judaica* | A | MED | | [64] |
| *Parietaria officinalis* | N | WE | | [82] |
| *Pipturus* sp. | N | HI | | [27] |
| *Urtica dioica* | N,A | F&S B&I WE ENA | ● | [36, 63, 72, 87] |
| *Urtica urens* | N | WE | | [35] |
| **Verbenaceae** | | | | |
| *Aloysia citrodora* | N | B&I | ●●● | |

*(Continued)*

**Table 1.** (Continued)

| Host species by family | Stage | Geographic area(s) | Cit. sci. obs. | Selected record source(s) |
|---|---|---|---|---|
| *Stachytarpheta* sp. | N | HI | | [127] |
| *Verbena bonariensis* | N | B&I | ●●● | |
| *Verbena hastata* | N | B&I | ●●● | |
| *Verbena litoralis* | N | HI | | [28] |
| *Verbena rigida* | N | WNA | | [29] |
| **Violaceae** | | | | |
| *Viola canina* | N | F&S B&I | ●● | [36] |
| *Viola epipsila* | N | F&S | | [36] |
| *Viola odorata* | N | WNA | | [29] |
| *Viola palustris* | N | M&NE | | [35] |
| *Viola riviniana* | N | B&I | ● | [37] |
| *Viola tricolor* | N | WE, ENA | | [5, 35] |
| *Viola x wittrockiana* | N | B&I | ●●● | |
| **Vitaceae** | | | | |
| *Parthenocissus henryana* | N | B&I | ●●● | |
| *Parthenocissus quinquefolia* | N | B&I | ●●● | |
| *Vitis vinifera* | N,A | WE EE MED | | [40, 78, 198–200] |
| **Zingiberaceae** | | | | |
| *Hedychium coronarium* | N | HI | | [27] |

Stage: N = nymph, A = adult, N,A = nymph and adult,? = unknown

Geographic area: F&S = Finland and Scandinavia, B&I = Britain and Ireland, WE = Western Europe, EE = Eastern Europe, MED = Mediterranean Basin, ENA = Eastern North America, WNA = Western North America, HI = Hawaii, NZ = New Zealand, AZ = Azores Islands,? = unknown. Western Europe includes France, Belgium, the Netherlands, Luxembourg, Germany, Switzerland, the Czech Republic, Austria and Slovenia. Eastern Europe includes the Baltic countries, Poland, Slovakia, Hungary, Serbia, Kosovo, North Macedonia, Romania, Bulgaria, Belarus, Ukraine, Moldova, Russia and Georgia. The Mediterranean Basin includes all countries bordering the Mediterranean except France and Slovenia (but, as a partial exception, includes Corsica). Eastern North America includes the USA and Canada east of the 100[th] meridian. Western North America includes the USA and Canada west of the 110[th] meridian. A number of Noury's [35] listings are from a source covering "Middle and Northern Europe", a category that does not fit our arbitrary divisions. These records are recorded as M&NE. A few North American references do not specify which section of the continent. These are recorded as EorWNA.

Citizen science observations: ●●● = new to the scientific record, ●● = new to B&I, ● = confirmation of earlier B&I record(s). Note that because they are host records new to science, there are no additional source citations for ●●● host species.

Sources: Published sources are cited by numbered references. Unpublished sources are cited as follows. Thirty unpublished observations by VT are referenced in the following format: VT-XX-YYYY-Z, where XX represents the standard two letter postal abbreviations for the American states, YYYY represents the year of observation, and Z represents the stage observed (N or A). Three unpublished observations by CH are referenced in an analogous format substituting "England" for the XX state abbreviation: CH-England-2019-N. Twenty-five unpublished nymphal observations provided by Saskia Hogenhout, Sam Mugford, Roberto Biello and Qun Liu (John Innes Centre) dating from 2019–2021 are referenced as HMBL-XX, where XX is England, Scotland or Italy. Thirty-three English nymphal records underlying the work of Jennifer Owen [38] and recovered from her archived papers are referenced simply as J. Owen. Five adult records are based on pin label information from the following collections: American Museum of Natural History (AMNH), University of California Riverside Entomology Research Museum (UCRC) and Snow Entomological Museum Collection (SEMC), listed by their respective abbreviations. Three nymphal observations from Scotland by John Raven (University of Dundee) and three from California by Richard Karban (University of California, Davis), all from 2022, are referenced as personal communications: JRpercom and RKpercom.

## Discussion

### *Philaenus spumarius* appears to be the most polyphagous insect herbivore

We start by comparing *P. spumarius* with other insects in terms of the number of host species exploited. At 1311 species, *P. spumarius* has, to our knowledge, more documented hosts than any other herbivorous insect. For comparison, Table 2 lists some of the serious contenders. Following Normark and Johnson [209], we limit comparisons to insects that feed directly on plant

**Table 2. Selected "extreme polyphage" insects, including the insect species with the highest documented numbers of host plants, and, for comparison, a few notorious examples, such the Spongy moth and the Spotted lantern fly, as well as the second ranking spittlebug, *Aphrophora alni*.** Note that *P. spumarius* has far more recorded hosts than any of the comparison species.

| Insect species | Order & Family | Common name | Number of host plants | | | Reference |
|---|---|---|---|---|---|---|
| | | | Species | Genera | Families | |
| *Philaenus spumarius* (L.) | Hemiptera: Aphrophoridae | Meadow spittlebug | 1311 | 631 | 117 | present work* |
| *Hyphantria cunea* (Drury) | Lepidoptera: Arctiidae | Fall webworm | 636 | 326 | 113 | [201]*† |
| *Coccus hesperidum* L. | Hemiptera: Coccidae | Brown soft scale | 552 | 417 | 138 | [50]* |
| *Epiphyas postvittana* (Walker) | Lepidoptera: Tortricidae | Light brown apple moth | 545 | 363 | 121 | [202] |
| *Aspidiotus nerii* Bouche | Hemiptera: Diaspididae | Ivy scale | 507 | 359 | 121 | [50]* |
| *Hemiberlesia lataniae* (Signoret) | Hemiptera: Diaspididae | Palm scale | 478 | 360 | 120 | [50]* |
| *Saissetia coffea* (Walker) | Hemiptera: Coccidae | Hemispherical scale | 412 | 313 | 112 | [50]* |
| *Lygus rugulipennis* Poppius | Hemiptera: Miridae | European tarnished plant bug | 402 | 226 | 57 | [49]* |
| *Lygus lineolaris* (Palisot de Beauvois) | Hemiptera: Miridae | Tarnished plant bug | 333 | 220 | 55 | [203]* |
| *Myzus persicae* (Sulzer) | Hemiptera: Aphididae | Green peach aphid | 305 | – | 72 | [204]‡ |
| *Helicoverpa armigera* (Hübner) | Lepidoptera: Noctuidae | Corn earworm | >300 | – | 68 | [205] |
| *Lymantria dispar* (L.) | Lepidoptera: Erebidae | Spongy moth | >300 | – | – | [206] |
| *Aphis fabae* Scopoli | Hemiptera: Aphididae | Blackbean aphid | 293 | – | 71 | [204]‡ |
| *Acherontia atropos* (L.) | Lepidoptera: Sphingidae | African death's-head moth | 208 | 102 | 40 | [207] |
| *Automeris postalbida* Schaus | Lepidoptera: Saturniidae | N/A | 188 | 125 | 59 | [207] |
| *Lycorma delicatula* (White) | Hemiptera: Fulgoridae | Spotted lantern fly | 172 | 100 | 33 | [208] |
| *Aphrophora alni* (Fallén) | Hemiptera: Aphrophoridae | Alder spittlebug | 145 | 89 | 38 | VT, unpublished* |

*Adjusted for generic redundancy.

†Includes some artificial cage feeding experiments.

‡UK hosts only

tissues. This excludes organisms like bees and syrphid flies that feed on pollen and nectar, flies that feed on rotting fruit, and leafcutter ants that attack plants but actually eat fungi that they cultivate on the plant material. None of the comparable insect species for which we have found data approach *P. spumarius* in number of host species. *Hyphantria cunea* (Drury), the fall webworm caterpillar weighs in closest, with a bit less than half the *P. spumarius* host numbers, though the webworm data include an unspecified number of artificial feeding tests [201], which we omitted in our compilation for *P. spumarius*. The closest arthropod competitor we have found is an arachnid, the red spider mite, *Tetranychus urticae* Koch, which is said to have more than 1100 hosts in over 140 plant families [210]. This puts *T. urticae* at the same order of magnitude as *P. spumarius* in host species number and substantially greater in host family number, the latter probably due to higher representation of plant families confined to the tropics.

## Why is *P. spumarius* so polyphagous?

Our results substantiate Ossiannilsson's undocumented 1981 assertion that *P. spumarius* has more than 1000 hosts [211], an informed guess that had become embedded in the literature (cf. [4, 54, 212]), despite a lack of supporting evidence. A 1977 statement by Halkka & Mikkola [213] that there are "nearly 4000 recorded food-plant species" is clearly a typographical error. The documented 1311 species in 117 families put *P. spumarius* squarely in the category of "extreme polyphage", defined by Normark & Johnson [209] as species that feed across more than 20 plant families. Like many extreme polyphages, it is a geographically widespread and invasive pest species, with very high population sizes. However, it does not exhibit other

**Table 3.** *Philaenus spumarius* **host plant families with 10 or more host species, ranked by number of host species.** Also included are the number of genera for each host family, the percent of host species from that family among all host species, the cumulative percent of all host species going down the ranking, the number of species in each host family, and an index of the occurrence of host species by family weighted for family size. The weighted occurrence index is the number of host species in each family divided by the total number of species in each family and expressed as a percentage. It provides a crude measure of the relative prominence of each host family, taking into account the large differences in numbers of species per family.

| Rank | Family | Number of host plant genera | Number of host plant species | Percent host species | Cumulative percent species | Number of species in family [48] | Weighted occurrence index |
|---|---|---|---|---|---|---|---|
| 1 | **Asteraceae** | 104 | 222 | 16.9% | 16.9% | 24700 | 0.90 |
| 2 | **Rosaceae** | 31 | 110 | 8.4% | 25.3% | 2950 | 3.73 |
| 3 | **Fabaceae** | 33 | 76 | 5.8% | 31.1% | 19500 | 0.39 |
| 4 | **Poaceae** | 49 | 73 | 5.6% | 36.7% | 12000 | 0.61 |
| 5 | **Lamiaceae** | 24 | 62 | 4.7% | 41.4% | 7530 | 0.82 |
| 6 | **Apiaceae** | 35 | 50 | 3.8% | 45.2% | 3575 | 1.40 |
| 7 | **Brassicaceae** | 27 | 43 | 3.3% | 48.5% | 3628 | 1.19 |
| 8 | **Caprifoliaceae** | 16 | 34 | 2.6% | 51.1% | 825 | 4.12 |
| 9 | **Caryophyllaceae** | 11 | 32 | 2.4% | 53.5% | 2625 | 1.22 |
| 10 | **Plantaginaceae** | 6 | 27 | 2.1% | 55.6% | 1900 | 1.42 |
| 11 | **Polygonaceae** | 8 | 26 | 2.0% | 57.6% | 1200 | 2.17 |
| 12 | **Ranunculaceae** | 10 | 24 | 1.8% | 59.4% | 2346 | 1.02 |
| 13 | **Onagraceae** | 5 | 21 | 1.6% | 61.0% | 656 | 3.20 |
| 14 | **Campanulaceae** | 5 | 19 | 1.4% | 62.5% | 2300 | 0.83 |
| 15 | **Ericaceae** | 10 | 19 | 1.4% | 63.9% | 4250 | 0.45 |
| 16 | **Primulaceae** | 4 | 19 | 1.4% | 65.4% | 2790 | 0.68 |
| 17 | **Boraginaceae** | 12 | 18 | 1.4% | 66.7% | 2535 | 0.71 |
| 18 | **Geraniaceae** | 3 | 18 | 1.4% | 68.1% | 830 | 2.17 |
| 19 | **Rubiaceae** | 8 | 18 | 1.4% | 69.5% | 13620 | 0.13 |
| 20 | **Asparagaceae** | 11 | 13 | 1.0% | 70.5% | 2900 | 0.45 |
| 21 | **Malvaceae** | 8 | 13 | 1.0% | 71.5% | 4225 | 0.31 |
| 22 | **Oleaceae** | 7 | 13 | 1.0% | 72.5% | 790 | 1.65 |
| 23 | **Betulaceae** | 5 | 12 | 0.9% | 73.4% | 167 | 7.19 |
| 24 | **Fagaceae** | 3 | 11 | 0.8% | 74.2% | 927 | 1.19 |
| 25 | **Papaveraceae** | 6 | 11 | 0.8% | 75.1% | 775 | 1.42 |
| 26 | **Pinaceae** | 3 | 11 | 0.8% | 75.9% | 228 | 4.82 |
| 27 | **Salicaceae** | 2 | 11 | 0.8% | 76.7% | 1220 | 0.90 |
| 28 | **Sapindaceae** | 3 | 11 | 0.8% | 77.6% | 1860 | 0.59 |
| 29 | **Solanaceae** | 6 | 11 | 0.8% | 78.4% | 2600 | 0.42 |
| 30 | **Amaryllidaceae** | 2 | 10 | 0.8% | 79.2% | 1600 | 0.63 |
| 31 | **Scrophulariaceae** | 6 | 10 | 0.8% | 79.9% | 1830 | 0.55 |

characteristics that Normark & Johnson [209] associate with extremely polyphagous insects, such as flightless females, larval dispersal, parthenogenesis or partiality to woody plants.

What characteristics have, in fact, contributed to the extraordinarily broad host range? Two factors are probably paramount. The first is xylem sap feeding, a nutritional mode that permits access to a food source that is similar across a wide range of host plants and not chemically defended (refs. in [214, 241]). Xylem feeding apparently permits *P. spumarius* to feed on almost any plant it can penetrate with its mouth parts. The second factor is wide geographical range and the ability to thrive in climates from Hawaii, just south of the Tropic of Cancer [28], to within 65 km of the Arctic Circle in Finland [36]. Most or all extreme polyphages have cosmopolitan or invasive distributions [209].

Among xylem feeding insects, which include spittlebugs, cicadas and one subfamily of leaf-hoppers, *P. spumarius* is singular in its occupation of most of the Holarctic plus multiple distant islands. By that standard, other xylem feeders have been modest travelers. In addition to *P. spumarius*, four other spittlebug species have been introduced from Europe to North America [215], two others to Hawaii [216], and one other to New Zealand [217]. Two xylem feeding leafhoppers have been introduced from North America to Europe [218, 219], and three cicada species have hopped from New Zealand's North Island to South Island ([220] & C. Simon, personal communication). None have achieved anything approaching the reach of *P. spumarius*. Perhaps not coincidentally, the other well-documented spittlebug extreme polyphage, *Aphrophora alni* (Table 2), is among the four other spittlebugs introduced from Europe to North America. It is not clear whether wide distribution is a cause or effect of extreme polyphagy. Each is clearly predisposed to promote the other [209].

*Philaenus spumarius* polyphagy seems to be a recent evolutionary development. It is one of a cluster of eight closely related *Philaenus* species living around the Mediterranean Basin [221]. Five are narrow monophages as nymphs, four feeding exclusively on the lily *Asphodelus ramosus* L. (or its close relatives *A. aestevus* or *A. microcarpus*) and one on *Eryngium* [222]. Two, *P. spumarius* and its very closely related sister species *Philaenus tesselatus*, are broad polyphages, though the extent of polyphagy is much less studied in *P. tesselatus* [223, 224]. The host status of the eighth species, *Philaenus arslani*, is uncertain. It has been collected from a modest variety of hosts, including three thistles, *Cistus* and "diverse shrubs" [150], all apparently but not explicitly adult hosts.

Maryańska-Nadachowska et al. [221, 225] propose that the line leading to *P. spumarius* originated from an *Asphodelus*-feeding ancestor between 7.9 and 3.7 Mya. If so, *P. spumarius* broke out of the Mediterranean monophage pack and spread to an enormous variety of hosts in a relatively short geological time period, exhibiting what Normark & Johnson [209] describe as a "niche explosion". Why? One answer might be the evolution of a more extensive arsenal of gene families involved in digestion, detoxification and transport of xenobiotics, as suggested for the red spider mite [210], the only arthropod we have found with a comparable host range, and for the green peach aphid [226] and corn earworm [205], among the runners up for most polyphagous insect herbivore (Table 2). On the other hand, the fact that xylem feeders encounter so few xenobiotics may make heroic detoxification capacity unnecessary. Ongoing work to sequence the complete *P. spumarius* genome [227] should provide data for a comparative analysis of the evolution of feeding versatility-related genes in relationship to niche explosion.

Whether or not accompanied by extensive changes in the food assimilation related genome, the rapid evolution of the *P. spumarius* line from narrow monophagy to extreme polyphagy may have been facilitated by the feeding ecology of the adults. *Asphodelus* lilies die back in the Mediterranean summer dry season. *Philaenus* species dependent on *Asphodelus* as nymphs move to alternative hosts as adults [222], typically ectomycorrhizal trees and shrubs [228], a broadening of host range that may have set the stage for the evolution of polyphagy in the *P. spumarius* line. Although it has been stated that Mediterranean climate *P. spumarius* aestivate on these summer hosts [44, 222], there is no evidence for a state of summer torpor or hibernation ([69] & VT observations in California).

The apparent evolution of an extreme polyphage from monophagic ancestors in a relatively short evolutionary interval is highly unusual. Polyphages are rare among herbivorous insects ([229, 230] and references therein), extreme polyphages even more so [209]. Had it been included in the most recent world survey of insect host plant breadth by Forister et al. [231], *Philaenus spumarius* would have been, in the most literal sense, off the charts. It seems to be one of a kind. It is also a clear counterexample to the suggestion that extreme polyphagy is an illusion based on multiple indistinguishable cryptic species feeding on different hosts [204,

226]. Extensive work on mitochondrial haplotype distribution in *P. spumarius* rules out multiple unrecognized cryptic species, although its mitochondrial lineages are bifurcated into two distinct clades [51, 52, 232] and one study suggested the presence of an unrecognized cryptic species in Anatolia and the Caucasus [233].

## Patterns in host plant usage

Given that *P. spumarius* seems able and willing to feed on almost any available host, what patterns in host usage can we discern? In sheer species numbers (Table 3) the Asteraceae (222) win hands down, with over twice as many hosts as the runner up Rosaceae (110), followed by the Fabaceae (76) and the Poaceae (73). The latter two high ranking groups merit special comment. Spittlebugs have a demonstrated affinity for nitrogen-fixing hosts, including many Fabaceae [214] (Fig 1a). In *P. spumarius* this is reflected in its pest status in legume forage crops in North America. Though it occurs on greater host species numbers in Asteraceae, it achieves highest densities on Fabaceae, up to 1280 nymphs/m$^2$ in on *M. sativa* [14]. The large numbers of Poaceae hosts (71) are surprising in the other direction. It has long been recognized that *P. spumarius* favors herbaceous dicots and is relatively rare on grasses [36]. While grasses as a group are not preferred hosts, the present results demonstrate that there are relatively large numbers of grass host species, which contribute markedly to total host diversity, and it is clear that *P. spumarius* is sometimes locally common on grasses. Booth [37], for example, found *P. spumarius* plentiful on grasses at some open sites in New Zealand and shaded sites in Wales, while Lester et al. [234] found *P. spumarius* to be relatively common on grasses during the Scottish professional BRIGIT survey.

At the high end of the host spectrum, *P. spumarius* is found not only on large numbers of Asteraceae species, but occurs in large numbers and high density on some individual species, including several *Solidago* spp. and a number of thistles. Among the Rosaceae, *Filipendula ulmaria* by itself accounted for 22% of 40,737 nymphal host records collected by Halkka and his colleagues in Finland [213]. This highlights a major limitation of species lists as a measure of host diversity. They count occurrence but not frequency, though local frequency is often recorded in the underlying sources.

Another way to look at host attraction is to compare the ratio of *P. spumarius* hosts in a given family to the total number of species in that family. Table 3 includes a weighted occurrence index, the percentage of *P. spumarius* host species among all species in a plant family. This corrects, in a rough and ready way, for the fact that some plant families are small and some are enormous. Given that *P. spumarius* has an essentially temperate distribution, however, it should be noted that the index will be highly conservative for plant families that have a large proportion of species in the tropics. Among families with at least ten *P. spumarius* hosts, the index ranges from a low of 0.12 for Rubiaceae to a high of 7.19 for Betulaceae. In relation to total species numbers, *P. spumarius* occurs on a small proportion of Rubiaceae and a high proportion of Betulaceae. The Pinaceae (4.82), Caprifoliaceae (4.00), Rosaceae (3.42) and Onagraceae (3.05) are also high scoring. In contrast, two of the top three host families, Asteraceae (0.86) and Fabaceae (0.38), are knocked out of this competition by their enormous species numbers. The high rankings of two families comprised solely of trees and shrubs, Betulaceae and Pinaceae, might seem counterintuitive for an insect that clearly favors herbs, but both groups are ectomycorrhizal and their high scores are consistent with the general overrepresentation of this category among spittlebug hosts [228].

Plant morphology also clearly plays an important role in host plant selection. Early instar nymphs are especially attracted to plants with rosette form or other forms of growth that favor closely apposed leaf surfaces [5], no doubt because compact leafy clusters in close proximity to

soil moisture form an advantageous early-instar nymphal microhabitat. Later instars tend to favor tall and robust perennial herbs [213]. On the other hand, plant features like abundant trichomes and lignification of tissues clearly deter *P. spumarius* feeding [90, 235]. When *P. spumarius* nymphs feed on woody plants it is invariably on new, unlignified growth, such as saplings, adventitious shoots of trees, or growing areas at the tips of branches ([5, 121] and our observations).

Are there any otherwise apparently suitable plants on which *P. spumarius* does not feed? The most intriguing possibility is crownvetch, *Coronilla varia* L. (synonym *Securigera varia* (L.) Lassen), a Eurasian legume widely planted for forage and roadside erosion control in Eastern North America. Wheeler [236] reports that he found *P. spumarius* "in small numbers as adults only" on crownvetch but excludes it from his extensive list of arthropods collected on crownvetch in Pennsylvania. He adds that F.V. Grau, the founder of crownvetch studies in the USA, reported that he had never seen spittlebugs on this species in 28 years of work. We found no other records for crownvetch, a particularly unexpected result because *C. varia* is a nitrogen-fixing forage legume, the category of host on which *P. spumarius* otherwise reaches greatest densities in the USA. This suggests, subject to experimental verification, that there may be something exceptional about its biology that repels *P. spumarius*. If so, it might be a candidate species for understory plantings in orchards and groves where there is a desire to suppress *P. spumarius* vector populations [237].

It also appears that *P. spumarius* nymphs may not occur on *Asclepias*, species of which are notoriously well-defended chemically. Beirne [65] reports that *P. spumarius* nymphs do not feed on *Asclepias* species and the only *P. spumarius Asclepias* record we have found is for adults in Maryland (Table 1). There are, however, *Asclepias* nymphal records for two *Lepyronia* species [126, 238], demonstrating that this genus is not off-limits to all spittlebugs. Schmidt [25] says that he never observed spittles on *Chenopodium* or *Atriplex*. *Chenopodium album*, the species to which he is most likely referring, has been widely recorded as a *P. spumarius* host, both nymphal and adult, but *Atriplex* has only been observed as a host once, and only for adults (Table 1), suggesting that this genus may not be hospitable to nymphs. In general, our results sustain the early observations of Schmidt [25] and Fabre [83] that nymphs feed successfully on many plants that are chemically well-defended.

Although *P. spumarius* has been recorded on several fern species (Table 1), it has not been recorded on bryophytes. Press and Whittaker [239] illustrate a spittle of the grass-feeding spittlebug *Neophilaenus lineatus* on a moss (*Polytrichum commune*), demonstrating that mosses are within the realm of plausible hosts. Notable plant categories on which *P. spumarius* records are rare in relationship to their numbers are Orchidaceae, Bromeliaceae, and CAM plants as a group. This is not surprising for orchids and bromeliads, the large majority of which are tropical epiphytes, putting them largely out of the geographical and ecological range of *P. spumarius*. It is more surprising for CAM plants, many in the Crassulaceae (eight species in Table 1), which are diverse and widely distributed in areas and habitats that *P. spumarius* frequents. CAM plants (which overlap to include many species in the Orchidaceae and Bromeliaceae) maintain close control of daytime transpiration, perhaps interfering with the accessibility of xylem sap.

## Implications for *Xylella fastidiosa* management

The most important lesson to be drawn from this review is that *P. spumarius* can and does feed on an extremely diverse array of plants, including, it appears, almost any vascular plant that comes its way with sufficiently accessible xylem vessels, the only apparent exceptions being, as noted above, *Coronilla varia* and *Asclepius* species. *Philaenus spumarius* nymphs are relatively

sessile, moving infrequently, if at all, among hosts. By analogy with their leafhopper vector counterparts [240], they probably lose any *X. fastidiosa* infection upon molting, which occurs five times during nymphal development. In consequence, nymphs are not effective vectors. In contrast, adults are known to be effective vectors [110] but their exact host range is much less certain, with many fewer hosts documented (Table 1), this in turn being substantially due to the difficulty noted in distinguishing between a functional host plant and one on which the insect is merely positioned. Although efficacy of transmission varies greatly with host species [1], *P. spumarius* adults are exceptionally well positioned to vector *X. fastidiosa* quickly and widely wherever the two co-occur, contingent on local conditions that favor the propagation of the bacterium within and among host plants [22, 40, 106]. Potential counter measures include: 1) local elimination or population reduction of *P. spumarius* by management of agriculturally adjacent host plants [2, 106, 212, 237], 2) reduction or elimination of plant sources of *Xylella fastidiosa* infection [21], and 3) reduction of target plant susceptibility, through selection of cultivars with genetic resistance or other measures that reduce plant vulnerability to transmission and/or infection [2, 21, 241, 242]. In turn, *P. spumarius* can be used as a sentinel organism to detect and monitor the presence of *X. fastidiosa* in local environments [243].

Going forward, we suggest several guidelines for future studies of *P. spumarius* on host plants. First and foremost, investigators should always specify life stage. A substantial number of past reports, especially from the agricultural sector, have omitted this important information. Second, to the degree possible, quantify the results. Counts are best, but even simple qualitative observations, such as "rare" or "abundant" are helpful, especially for observations on adults. Third, where *P. spumarius* nymphs are abundant, record the plants on which they are apparently absent, especially in instances in which the uninfested plant species are frequent and apparently suitable for feeding. This will assist in the ongoing search for alternative understory plants suitable for reducing *P. spumarius* numbers in agricultural settings. Recent screening of nymphs in the Basilicata Region of Italy by Trotta et al. [11] is a model for this approach. The authors include data on 48 plant species that hosted nymphs *and* on 17 species that did not.

## Lessons from the BRIGIT citizen scientist project

Mass-participation citizen science projects have a well-established history of contributing environmental and ecological data over numerical, spatial and temporal scales that would be impossible to collect by professional researchers alone [244, 245]. Concerns about the quality of such data [246] are counterbalanced by recent studies indicating that the reliability of citizen science data can be significantly enhanced by training, data validation and ground-truthing [247, 248].

The BRIGIT citizen science project proved to be a powerful tool for the rapid collection of large amounts of *P. spumarius* host plant usage data across the UK. Spittle was found to be a reliable focus for citizen science activity, being highly visible during the nymphal season and largely unmistakable for any other natural phenomena. At least in the UK, therefore, spittle on herbaceous dicots could be used for rapid assessment of both the presence and relative abundance of *P. spumarius*. Such estimates could form part of a future surveillance strategy for *X. fastidiosa*, although many other factors would need to be considered when assessing the risk of bacterial transmission.

Finally, we note that the BRIGIT project is the latest installment in a long history of amateur contributions to *P. spumarius* host records, including the major works of Hugo Schmidt [25, 26], Ernest Noury [35] and Jennifer Owen [38]. In that sense, the work reported here is a monument to synergism between professional scientists and dedicated amateurs in the advancement of knowledge.

## Acknowledgments

We dedicate this paper to our departed colleagues Olli Halkka and David Lees, and to John Whittaker, pioneers in evolutionary and ecological studies of *P. spumarius* and other spittlebugs. We thank librarians Mary Beth Riedner (Roosevelt University, retired), Gwen Short (Ohio State University) and Mai Reitmeyer (American Museum of Natural History) for tracking down some of the more obscure and hard-to-access references necessary for a review of this kind. Adeline Soulier-Perkins (Museum National d'Histoire Naturelle) provided a hard-to-find copy of Noury's host compendium [35]. Martin Harvey (UK Biological Records Centre), Kirsty Gamble (Leicestershire and Rutland Environmental Records Centre) and Alison Clague (Leicester County Council) helped us track down Jennifer Owen's archived host records. Chris Simon (University of Connecticut) provided unpublished information on New Zealand cicada introductions. Benjamin Normark (University of Massachusetts, Amherst) provided helpful insights and leads on extreme polyphages. Numerous colleagues over the years have shared unpublished host observations, a handful used here and all valuable to the host tracking effort. The BRIGIT project was funded by UK Research and Innovation through the Strategic Priorities Fund, by a grant from the Biotechnology and Biological Sciences Research Council, with support from the UK Department for Environment, Food and Rural Affairs and the Scottish Government, to AJAS and CH. We thank Saskia Hogenhout, Sam Mugford, Roberto Biello and Qun Liu (John Innes Centre, UK) for providing the worldwide host plant records collected during the BRIGIT project.

## Author Contributions

**Conceptualization:** Vinton Thompson, Alan J. A. Stewart.

**Data curation:** Vinton Thompson, Claire Harkin, Alan J. A. Stewart.

**Formal analysis:** Alan J. A. Stewart.

**Funding acquisition:** Alan J. A. Stewart.

**Investigation:** Vinton Thompson, Claire Harkin, Alan J. A. Stewart.

**Methodology:** Alan J. A. Stewart.

**Validation:** Vinton Thompson, Claire Harkin, Alan J. A. Stewart.

**Writing – original draft:** Vinton Thompson.

**Writing – review & editing:** Vinton Thompson, Claire Harkin, Alan J. A. Stewart.

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
