## [Decision Letter · Decision Letter 0]

7 Aug 2023

PONE-D-23-20831The most polyphagous insect
herbivore? Host plant associations of the Meadow spittlebug, Philaenus spumarius
(L.)PLOS ONE

Dear Dr. Stewart,

Thank you for submitting your manuscript to PLOS ONE. After careful consideration, we
feel that it has merit but does not fully meet PLOS ONE’s publication criteria as it
currently stands. Therefore, we invite you to submit a revised version of the
manuscript that addresses the points raised during the review process.

Please submit your revised manuscript by Sep 21 2023 11:59PM. If you will need more
time than this to complete your revisions, please reply to this message or contact
the journal office at plosone@plos.org. When
you're ready to submit your revision, log on to https://www.editorialmanager.com/pone/ and select the 'Submissions
Needing Revision' folder to locate your manuscript file.

Please include the following items when submitting your revised
manuscript:A rebuttal letter that responds to each point raised by the academic
editor and reviewer(s). You should upload this letter as a separate file
labeled 'Response to Reviewers'.A marked-up copy of your manuscript that highlights changes made to the
original version. You should upload this as a separate file labeled
'Revised Manuscript with Track Changes'.An unmarked version of your revised paper without tracked changes. You
should upload this as a separate file labeled 'Manuscript'.If you would like to make changes to your financial disclosure,
please include your updated statement in your cover letter. Guidelines for
resubmitting your figure files are available below the reviewer comments at the end
of this letter.

We look forward to receiving your revised manuscript.

Kind regards,

Dr. Janice L. Bossart

Academic Editor

PLOS ONE

Journal Requirements:

Additional Editor Comments:Both reviewers feel your manuscript will
have significant impact in the scientific community, especially for researchers
dealing with *Xylella fastidiosa* impacts. There concerns were
generally minor and shouldn't require much time to
address. Please note that
PLoS ONE doesn't rely on copy editors. Please carefully scrutinize your manuscript
for typos and grammatical errors, and double check that your references are all
inclusive, all cited within the text, and all are formatted the
same.

Reviewers' comments:

Reviewer's Responses to Questions

**Comments to the Author**

1. Is the manuscript technically sound, and do the data support the conclusions?

Reviewer #1: Yes

Reviewer #2: Yes

2. Has the statistical analysis been performed
appropriately and rigorously? 

Reviewer #1: N/A

Reviewer #2: N/A

3. Have the authors made all data underlying the
findings in their manuscript fully available?

Reviewer #1: Yes

Reviewer #2: Yes

4. Is the manuscript presented in an intelligible
fashion and written in standard English?

Reviewer #1: Yes

Reviewer #2: Yes

5. Review Comments to the Author

Reviewer #1: The manuscript is a comprehensive review od the meadow spittlebug host
plants. I have just minor comments and suggestions the authors will find in the
attached pdf.

In addition:

1) I think it'd be better to present table 1 as supplementary material

2) I suggest to remove the conclusions on the importance of citizen science in
surveys on Xylella fastidiosa vectors. The bacterium transmission dynamic is indeed
far more complicated than the "simple" assessment of the number of spittles in a
certain area/orchard, and these kind of statements could be easily misinterpreted by
both the scientific community and the stakeholders.

Reviewer #2: The authors carried out a quite impressive research on the host plant
range of the spittlebug Philaenus spumarius. They both delved the, often ancient,
literature but provided also the results of an interesting citizen science project
on the topic embedded in the EU project BRIGIT.

The results are interesting and useful, especially for the many scientists studying
the insect as the key vector of Xylella fastidiosa in Europe. The impressive list of
host plants of the extreme polyphagous insect topic of this article surely may
contain some mistakes or imprecisions, but it try to make order in a otherwise
cluttered amount of literature and information, some really old and not easily
accessible even to the average expert.

I have only few little corrections and considerations.

#1

The MS provides some tables but lacks graphs. There is not a graphical way to
condense some of the information present in table 2 or 3 in a plot and maybe place
the table as a supplementary material? It could be more immediate to the reader the
importance of some plant family as host plants or the comparison between P.
spumarius and other extreme polyphagous insects. This is just a suggestion to
increase the readability of the paper and it is not compulsory.

l. 145-146: “with which VT is familiar”. I found this expression a little too
personal and not really supported by data. It would be better to cite some
literature on California climate or briefly describe which climate type the authors
referred to. In any case this clarification is in my opinion not necessary and can
be avoided.

l. 305: “fantastic”. Although I agree that P. spumarius is a fantastic insect this
expression is maybe a little too personal in my opinion.

Table 3: “weighted occurrence index”. Although the authors state that this is a rough
way to account for differences in number of species per family, it would be ideal if
the number of species/family would be weighted for the species per family present in
the area of distribution of P. spumarius and not worldwide. Is it feasible to do so
or is it too much work and beyond the scope of this MS? Think about that.

l. 347: I am not a English native speaker, and I had never read this expression
before. Consider if it is really needed a similar colloquial English expression.

l. 417: delete one “of”.

l. 468: correct Rubiaciae in Rubiaceae.

l. 495-96: I have to say that I am a little skeptical about the reports on the
apparent avoidance of Securigera varia by P. spumarius nymphs. I have personally
found spittlebug nymphal stages (not many) on closely related species and I find
difficult to think that this particular species is anything really special for P.
spumarius. I just suggest to be more careful in presenting this plant as a possible
candidate for suppressing the population of the insect pest, also because the study
of Morente et al. cited used very different plants from other taxonomic families as
trap crops. More dedicated studies are needed before planting a Fabaceae plant to
suppress spittlebug population in a X. fastidiosa susceptible crop orchard!

l. 527: statement not really supported, we do not know well the host range of adults,
given also the underlined difficulty in assessing a real host plant, i.e., a plant
on which they feed. I would suggest to stress the lack of information on adult stage
host plant range and not assume that it is comparable to the one of nymphal
stages.

6. PLOS authors have the option to publish the peer
review history of their article (what does this mean?). If published, this will
include your full peer review and any attached files.

If you choose “no”, your identity will remain anonymous but your review may still be
made public.

**Do you want your identity to be public for this peer review?** For
information about this choice, including consent withdrawal, please see our
Privacy Policy.

Reviewer #1: No

Reviewer #2: No

---

## [Author Response · Author response to Decision Letter 0]

29 Aug 2023

Dr. Janice L. Bossart

Academic Editor

PLOS ONE

Dear Dr Bossart

Thank you for considering our submission to PLOS ONE and for the perceptive comments
and constructive suggestions for improvement of the manuscript. Our responses to
these comments are listed below. We have reproduced in bold the comments from the
journal; our response to each one then follows in normal font. Line numbers in our
responses refer to the revised manuscript without track changes. 

Journal Requirements:

1. Journal style requirements

We have reviewed the style requirements and are satisfied that the manuscript meets
them.

2. Data Availability statement

Our entire dataset is contained within Table 1.

3. Reference list

We have reviewed the reference list and are satisfied that it is complete, correct
and meets the formatting requirements.

Additional Editor Comments:

Both reviewers feel your manuscript will have significant impact in the scientific
community, especially for researchers dealing with Xylella fastidiosa impacts. There
concerns were generally minor and shouldn't require much time to address.

Thank you for this endorsement of our work.

Comments to the Author

Reviewer #1: The manuscript is a comprehensive review of the meadow spittlebug host
plants. I have just minor comments and suggestions the authors will find in the
attached pdf.

Thank you for this assessment.

In addition:

1) I think it'd be better to present table 1 as supplementary material

We note the reviewer’s suggestion, but we feel strongly that Table 1 belongs, as
currently positioned, in the main body of the paper. Our work is unusual, in that
its main purpose is to bring directly to the scientific community an annotated,
complete compilation of Philaenus spumarius host plants. The primary target audience
is the large and growing contingent of scientists and agronomists working with the
plant pathogen Xylella fastidiosa. The full table of P. spumarius hosts will become
a staple of their work and they will consult it frequently, most often searching for
host plants that they have observed in association with P. spumarius in natural or
agricultural habitats where P. spumarius is a vector or potential vector. Their
needs will be best served by having the table visible, readily accessible and at
hand, with no need to call up supplementary materials separate from the pdf or html
versions of the primary work. We note that the European Food Safety Authority (EFSA)
publishes a comparable compilation of X. fastidiosa host plants. The tables for
their compilation of several hundred Xylella host plant species are published in
full in the main body of the text and heavily consulted by workers in
Xylella-related studies.

We understand that the table is long and would be impractical to print as part of a
traditional print-based journal article. That was a primary motivation for
submitting to PLOS ONE. In a solely on-line venue, issues of page length are not a
barrier to presenting detailed data sets in full where they are central to the story
and to the interests of the target audience. For our readers, we would like to take
full advantage of this flexibility and present the core of our work, and the part of
it most likely to be accessed regularly, in the most visible and accessible way.

2) I suggest to remove the conclusions on the importance of citizen science in
surveys on Xylella fastidiosa vectors. The bacterium transmission dynamic is indeed
far more complicated than the "simple" assessment of the number of spittles in a
certain area/orchard, and these kind of statements could be easily misinterpreted by
both the scientific community and the stakeholders.

We completely agree that the number of spittles is only one component in assessing
Xylella transmission risk. However, this was only two sentences at the end of this
section and we feel that it would be useful to retain the rest of the text, even
though the reviewer has suggested deleting the whole section. Accordingly, we have
modified the final two sentences as follows:

At least in the UK, therefore, spittle on herbaceous dicots could be used for rapid
assessment of both the presence and relative abundance of P. spumarius. Such
estimates could form part of a future surveillance strategy for X. fastidiosa,
although many other factors would need to be considered when assessing the risk of
bacterial transmission.

Reviewer #2: The authors carried out a quite impressive research on the host plant
range of the spittlebug Philaenus spumarius. They both delved the, often ancient,
literature but provided also the results of an interesting citizen science project
on the topic embedded in the EU project BRIGIT. The results are interesting and
useful, especially for the many scientists studying the insect as the key vector of
Xylella fastidiosa in Europe. The impressive list of host plants of the extreme
polyphagous insect topic of this article surely may contain some mistakes or
imprecisions, but it try to make order in a otherwise cluttered amount of literature
and information, some really old and not easily accessible even to the average
expert.

Thank you for these positive comments.

I have only few little corrections and considerations.

#1

The MS provides some tables but lacks graphs. There is not a graphical way to
condense some of the information present in table 2 or 3 in a plot and maybe place
the table as a supplementary material? It could be more immediate to the reader the
importance of some plant family as host plants or the comparison between P.
spumarius and other extreme polyphagous insects. This is just a suggestion to
increase the readability of the paper and it is not compulsory.

We have experimented with various graphical representations of these two tables but
have concluded that the data are best portrayed in tabular format. 

l. 145-146: “with which VT is familiar”. I found this expression a little too
personal and not really supported by data. It would be better to cite some
literature on California climate or briefly describe which climate type the authors
referred to. In any case this clarification is in my opinion not necessary and can
be avoided.

We accept the point about this expression. We contend, however, that some
clarification of whether the observations made by DeLong & Severin referred to
nymphs or adults is useful. Accordingly, we have deleted the sentence in question
and replaced it with the following (lines 146-151):

All were made in Alameda County, on the sunnier, warmer side of San Francisco Bay, in
April or the first three weeks of May, or in San Francisco, on the foggier, cooler
side of the bay, in April, May or the first week of July. In these areas, for the
periods in question, most or all P. spumarius individuals are still in the nymphal
stage (VT observations) and it may be reasonably inferred that DeLong & Severin
observed nymphs for all listed hosts.

l. 305: “fantastic”. Although I agree that P. spumarius is a fantastic insect this
expression is maybe a little too personal in my opinion.

This is a fair point. We have replaced ‘fantastic’ with ‘extraordinary’ (line
308).

Table 3: “weighted occurrence index”. Although the authors state that this is a rough
way to account for differences in number of species per family, it would be ideal if
the number of species/family would be weighted for the species per family present in
the area of distribution of P. spumarius and not worldwide. Is it feasible to do so
or is it too much work and beyond the scope of this MS? Think about that.

Thank you for this interesting point. It is true that, ideally, the summation of
plant species in each family should include only those that overlap with the actual
distribution of P. spumarius. However, such an analysis would require extracting the
geographical distributions of every species in the 117 plant families concerned.
Apart from the work involved in such an exercise, the ranges of some of these may
not be known with sufficient spatial precision to make this a worthwhile endeavour.
Accordingly, we have inserted the following sentence into the text on lines
469-471:

Given that P. spumarius has an essentially temperate distribution, however, it should
be noted that the index will be highly conservative for plant families that have a
large proportion of species in the tropics.

l. 347: I am not a English native speaker, and I had never read this expression
before. Consider if it is really needed a similar colloquial English expression.

This is a perfectly valid point. We have replaced the sentence with the following one
(line 350):

We start by comparing P. spumarius with other insects in terms of the number of host
species exploited. 

l. 417: delete one “of”.

Done (line 450).

l. 468: correct Rubiaciae in Rubiaceae.

Done (line 473).

l. 495-96: I have to say that I am a little skeptical about the reports on the
apparent avoidance of Securigera varia by P. spumarius nymphs. I have personally
found spittlebug nymphal stages (not many) on closely related species and I find
difficult to think that this particular species is anything really special for P.
spumarius. I just suggest to be more careful in presenting this plant as a possible
candidate for suppressing the population of the insect pest, also because the study
of Morente et al. cited used very different plants from other taxonomic families as
trap crops. More dedicated studies are needed before planting a Fabaceae plant to
suppress spittlebug population in a X. fastidiosa susceptible crop orchard!

This is a valid point given the current lack of direct experimental evidence on the
susceptibility of this plant species to spittlebug feeding. We have modified the
text to be more cautious, as follows (lines 499-502):

This suggests, subject to experimental verification, that there may be something
exceptional about its biology that repels P. spumarius. If so, it might be a
candidate species for understory plantings in orchards and groves where there is a
desire to suppress P. spumarius vector populations [237].

l. 527: statement not really supported, we do not know well the host range of adults,
given also the underlined difficulty in assessing a real host plant, i.e., a plant
on which they feed. I would suggest to stress the lack of information on adult stage
host plant range and not assume that it is comparable to the one of nymphal
stages.

This is an important point; thank you. We have modified the sentence as follows
(lines 532-536):

In contrast, adults are known to be effective vectors [110] but their exact host
range is much less certain, with many fewer hosts documented (Table 1), this in turn
substantially due to the difficulty noted in distinguishing between a functional
host plant and one on which the insect is merely positioned.

Responses to in-text comments by Reviewer #1 (line numbers refer to PDF file with
reviewer comments). Please note that all suggestions for minor single-word
adjustments or typographic errors have been accepted.

L 21: Please summarize the main goal of the present work and the methodology

To provide a broader context, we have inserted the following sentence at the start
(lines 23-24):

A comprehensive list of all known host plant species utilised by the Meadow
Spittlebug (Philaenus spumarius (L.)) is presented, compiled from published and
unpublished sources.

L 33-36: Unclear, please rephrase

We assume the reviewer is referring to the word ‘captured’ (line 36). We have now
replaced this with ‘recorded’ which we suggest clarifies the meaning.

L 46-7: Remove, it's not consistent with the statements below (that there was a
proliferation of studies on spittlebug hosts upon Xylella first outbreak)

The key concept here is ‘comprehensive review’. We have re-phrased the next sentence
as follows (replacement words underlined, lines 49-52):

Concern with X. fastidiosa has led to a proliferation of recent studies adding new P.
spumarius hosts (e.g. [6–11]), including the BRIGIT citizen scientist initiative in
Britain that enlisted amateurs to identify P. spumarius host plants [12].

L 171: Please explain what do you mean by "problematic"

We are referring to records where the species identification may be in question. We
have replaced ‘problematic’ with ‘unreliable’.

L 211-2: Please rephrase.

We are unclear why this sentence is problematic. Nevertheless, we have added the word
‘representing’ to make the sentence clearer (line 216):

We also recognize the possibility that some of the original plant identifications may
have been in error, representing another, hopefully small, source of noise in the
data.

Table 3: How can it be >1? (referring to the weighted occurrence index) This
comment is repeated for lines 367-9.

As explained in the table’s legend and on lines 472-3, the weighted occurrence index
is a percentage and can therefore assume values greater than one. 

L 376: Please add also Avosani, S., Nieri, R., Mazzoni, V., Anfora, G., Hamouche, Z.,
Zippari, C., ... & Cornara, D. (2023). Intruding into a conversation: how
behavioral manipulation could support management of Xylella fastidiosa and its
insect vectors. Journal of Pest Science, 1-17.

We have added this reference to the in-text citation (line 379).

Thank you once again for considering our paper for publication in PLOS ONE. We look
forward to receiving your decision on the revised version.

Yours sincerely,

Alan Stewart

Claire Harkin

Vinton Thompson

to Reviewers.docx
---

## [Editor Report · Decision Letter 1]

4 Sep 2023

The most polyphagous insect herbivore? Host plant associations of the Meadow
spittlebug, Philaenus spumarius (L.)

PONE-D-23-20831R1

Dear Dr. Stewart,

We’re pleased to inform you that your manuscript has been judged scientifically
suitable for publication and will be formally accepted for publication once it meets
all outstanding technical requirements.  Congratulations and thank you for your
efforts to address reviewer concerns.  Although I realize it would have taken
additional effort, I do think a weighted occurrence index based only on plant
species that overlap with the spittlebug's distribution would have significantly
increased the value of your manuscript.

Kind regards,

Dr. Janice L. Bossart

Academic Editor

PLOS ONE

Additional Editor Comments (optional):

Please be sure the formatting of your manuscript follows PLOS ONE guidelines. The
format on the uploaded pdf document I viewed was atypical.
---

## [Editor Report · Acceptance letter]

13 Sep 2023

PONE-D-23-20831R1 

The most polyphagous insect herbivore? Host plant associations of the Meadow
spittlebug, *Philaenus spumarius* (L.) 

Dear Dr. Stewart:

I'm pleased to inform you that your manuscript has been deemed suitable for
publication in PLOS ONE. Congratulations! Your manuscript is now with our production
department. 

Kind regards, 

on behalf of

Dr. Janice L. Bossart 

Academic Editor

PLOS ONE